# Spatial Differentiation of Carbon Budgets and Carbon Balance Zoning in China Based on the Land Use Perspective

**Hui Wen, Yi Li, Zirong Li, Xiaoxue Cai and Fengxia Wang ***

School of Tourism, Hainan University, Haikou 570228, China
* Correspondence: summer_wangfx@126.com; Tel.: +86-189-7621-5081

**Abstract:** Carbon emission reduction in China is of great significance to curb global warming. Based on the land use perspective, the spatial characteristics of carbon emissions and carbon sinks in 30 Chinese provinces were analyzed and the carbon balance was partitioned by combining the economic contribution coefficient (ECC) and ecological support coefficient (ESC). It was found that (1) the intensity of land use in China is strong, being high in the southeast and low in the northwest, divided by the Heihe–Tengchong Line, and carbon source and carbon sink lands show clear spatial heterogeneity. (2) Total carbon emissions show clear spatial heterogeneity. Carbon emissions from construction land are the main source of carbon emissions. The carbon emission intensity and per capita carbon emissions are both high in the north and low in the south. (3) The total carbon sink is high in the north–south and low in the middle of the country, and woodland and grassland are the main sites of terrestrial carbon absorption. The overall carbon sink intensity shows a continuous decrease from southeast to northwest. (4) Based on the ECC and ESC indicators, 30 provinces were divided into four carbon zones and differentiated low-carbon development suggestions are proposed.

**Keywords:** land use; carbon emissions; carbon budget; carbon balance; low carbon development





## 1. Introduction

Global warming caused by the rapid increase in carbon emissions has become a major focus of attention, and the promotion of green and low-carbon economic development has become the consensus strategy within the international community to address and improve climate change [1]. Therefore, the milestone international legal text, the Paris Agreement, has been successfully implemented, making arrangements for global action against climate change after 2020 and forming a new pattern of global climate governance. China became the world's top carbon emitter in 2009 and accounted for 23.87% of the world's total carbon emissions in 2017 [2]. In order to strengthen China's Nationally Determined Contributions and implement the commitment to the Paris Agreement, China has pledged to strive to achieve a carbon peak by 2030 and carbon neutrality by 2060. As a basic factor of human production, land hosts human socioeconomic activities and is also an important carbon source and sink [3]. Land use/cover change alters the original land cover pattern and ecosystem structure and the processes and functions of terrestrial ecosystems, directly affecting terrestrial carbon cycling and energy flow [4,5]. Since the start of reform and opening up, China has experienced rapid economic development, industrialization, and urbanization, and the rapid expansion of land for construction has led to a decrease in land for carbon sinks and a continuous increase in energy consumption and carbon emissions [6]. In recent years, China has actively promoted energy conservation and the construction of an ecological civilization, the establishment of a series of protective forests, and the implementation of the project of returning farmland to forests and grasses, which have continuously improved the total quantity and quality of forest and grass resources, but determining the method to achieve low-carbon cyclic development while maintaining economic growth is still a serious challenge for China [7]. The intensity of land

development and energy consumption varies greatly among Chinese provinces; therefore, scientific and accurate measurement and analysis of total carbon emissions and carbon sinks in different provinces is key to formulating carbon emission reduction regulation and control policies that meet the characteristics of each province and reasonably promote low-carbon development in each region.

The impact of land use change on carbon emissions and sequestration is a trending research topic. There have been a large number of studies on the relationship between land use change and carbon balance, which are mainly divided into the following categories: First, studies on carbon source/sink estimation for a single land use type, such as forest [8], grassland [9], cropland [10], and construction land [11]; however, a single land use type cannot reflect the wholeness of a complex ecosystem, for which there are fewer studies. Second, the carbon sources/sinks of several terrestrial ecosystems are estimated from different perspectives and geographical scales, and their carbon balance status and spatial and temporal evolution are analyzed. At the national scale, land-use carbon emissions are characterized by high emissions in the east and low emissions in the west of China, mainly from construction land [12]. At the regional scale, the urban clusters are used as the study areas, with investigations on the characteristics of the carbon sink and source distribution and carbon balance in the Beijing–Tianjin–Hebei region [13] and on the characteristics of the carbon emission distribution in the Yellow River region [14]. At the provincial and city/county scales, scholars have studied carbon emissions in Jiangsu [15], Hubei [16], Chengdu [3], and Zichang County [17], which were used as the study areas. Third, scholars have conducted a great deal of research around carbon balance, zoning, and optimization schemes, and carbon offsetting has become a new field arising from the context of global climate change and green low-carbon development. Xia et al. classified nine types of spatial carbon compensation optimization zones in the Beijing–Tianjin–Hebei city cluster by the SOM-K-means model [18], and Zhao et al. classified the Central Plains Economic Zone into five types of zones based on carbon balance zoning theory, namely, the carbon intensity control zone, carbon balance zone, carbon sink function zone, total carbon control zone, and low carbon optimization zone [19].

In general, there have been systematic studies on carbon emissions, carbon sinks, and the land use carbon balance, which are important references for this work [8,12,13,19]; however, the following problems are still worth exploring in depth. First, the subdivision of land use types is not deep enough, which leads to large errors in the measurement results of the land use carbon budget. Second, most of the carbon emission factors for energy consumption use data published by the Intergovernmental Panel on Climate Change; however, China, as a large energy-consuming country, needs to determine its carbon emission factors through field measurements. Third, most scholars analyze the land use carbon budget from two perspectives, total carbon emissions and total carbon sinks, but fail to study it from multiple perspectives, such as carbon emission intensity, per capita carbon emissions, and carbon sink intensity. Finally, the existing studies focus on carbon balance zoning based on carbon budget accounting, without further considering the economic contribution of carbon emissions and the ecological role of carbon sinks.

Therefore, to fulfil the aforementioned gaps, this study analyzes the land use status and intensity of each Chinese province to understand the pattern of land use carbon sources/sinks as a whole. Secondly, it measures the total amount of carbon emissions and carbon sinks and analyzes their spatial distribution. Finally, it integrates the ECC and ESC of carbon emissions into the carbon balance zoning of the country and differentiated low-carbon optimization suggestions are put forward according to local conditions, aiming to provide a theoretical basis for relevant departments to adjust land use policies and improve the distribution efficiency of carbon emission reduction tasks in each province. The research framework is shown in Figure 1.

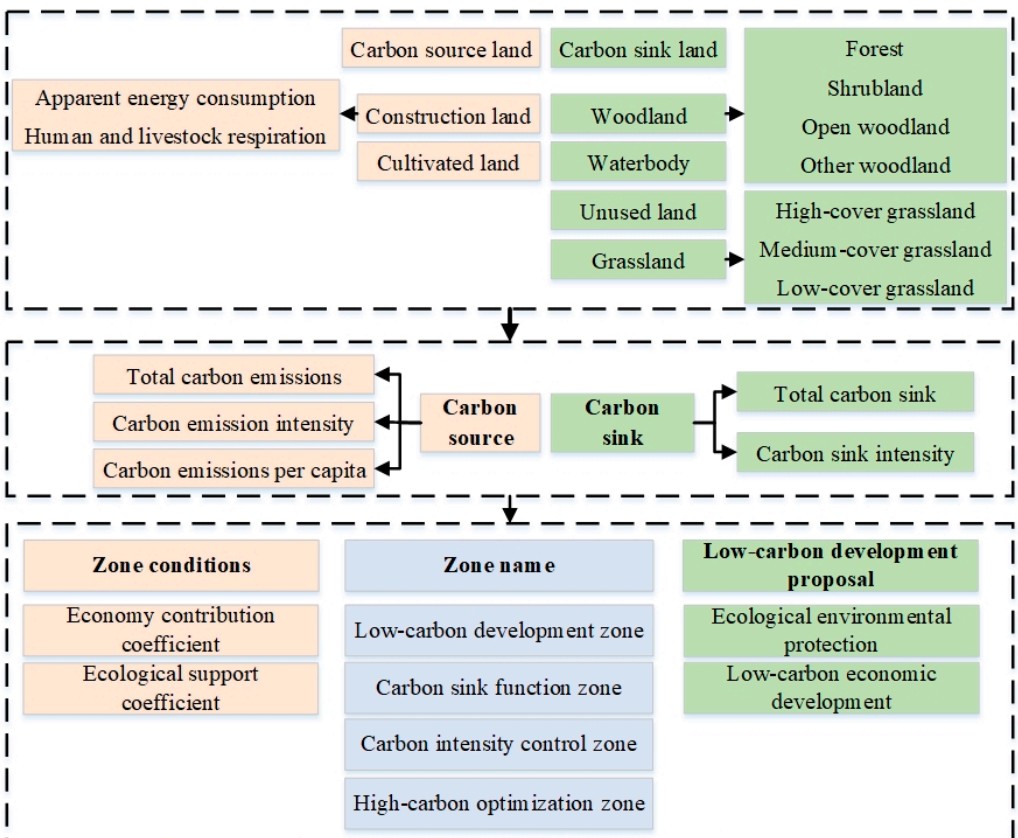

**Figure 1.** Research framework.

The innovative values and research contributions of this study are as follows: first, this study will improve the subdivision of land use types to account for land use carbon emissions and carbon sinks more accurately in China. Woodlands are subdivided into forest, shrubland, open woodland, and other woodland, and grasslands are subdivided into high-cover grassland, medium-cover grassland, and low-cover grassland. Second, carbon emissions from energy consumption are quoted with a carbon emission coefficient that is more in line with China's actual situation, which is derived from field measurements of 602 coal samples from the 100 largest coal mining areas in China. Third, this study analyzes the land use carbon balance through multiple indicators of total carbon emissions, carbon emission intensity, per capita carbon emissions, total carbon sink, and carbon sink intensity, and it uses the spatial autocorrelation model to reveal the spatial differentiation characteristics of the carbon balance more deeply. Finally, considering the economic contribution coefficient (ECC) and ecological support coefficient (ESC), from the perspective of carbon balance zoning, it divides the country into four zones: A low-carbon development zone, a carbon sink function zone, a carbon intensity control zone, and a high-carbon optimization zone, and the low carbon development path of each province is proposed in a targeted manner.

## 2. Data Sources and Methods

### 2.1. Data Sources and Preparation

The data used in this study mainly include land use, energy consumption, socioeconomic, and carbon emission and carbon sequestration coefficients data of 30 provincial-level administrative regions in China (excluding Tibet, Hong Kong, Macao, and Taiwan). (1) Land use data: The land use data of each province in China in 2018 were obtained from the Resource and Environment Science and Data Center of the Chinese Academy of Sciences with a spatial resolution of 1 km and an overall accuracy of more than 93% [20].

Land use types were divided into the following categories: Cultivated land, woodland, grassland, waterbody, construction land, and unused land, among which woodland was subdivided into forest, shrubland, open woodland, and other woodland, and grassland was subdivided into high-cover grassland, medium-cover grassland, and low-cover grassland. Therefore, land use types were divided into 11 categories in total. (2) Energy consumption data: The carbon emission data of energy consumption came from the provincial carbon emission inventory of Carbon Emission Accounts and Datasets for Emerging Economies, which is based on apparent energy consumption (raw coal, crude oil, and natural gas), and its emission factors are more in line with the Chinese reality than previous datasets with higher accuracy by measuring 602 coal samples from the 100 largest coal mining areas in China [21]. (3) Socioeconomic data: GDP, year-end total population, and livestock population (pigs, cattle) by province were acquired from *China statistical yearbook* [22]; the country was divided into four regions—Eastern, Northeastern, Western, and Central—according to the *China statistical yearbook* (Figure 2). (4) Carbon emission and carbon sequestration coefficients were derived from the results of published studies (Table 1).

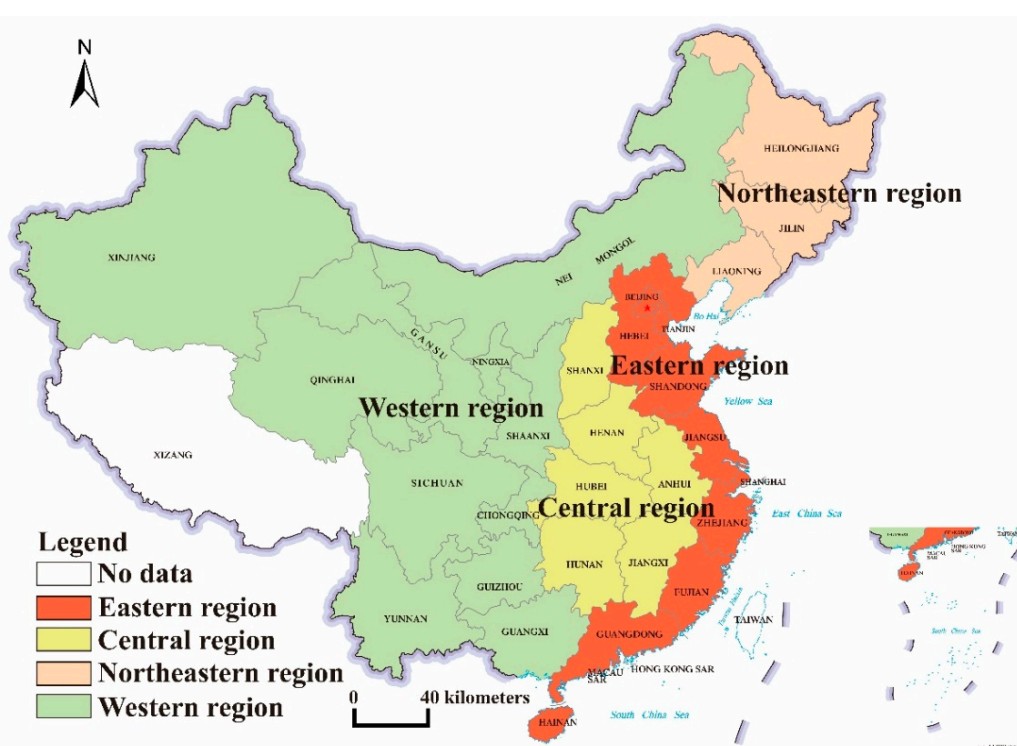

**Figure 2.** Four regions' geographical divisions of China. Note: This map is based on the standard map with the review number GS (2019)1684 downloaded from the Ministry of Natural Resources and produced using ArcGIS software, with no modification to the base map, the same below.

**Table 1.** Carbon emission and sequestration coefficients.

| Notation | Carbon Emission Component | Coefficient | Units | Source |
|---|---|---|---|---|
| $K_1$ | Raw coal | 0.499 | kg C/kg | Liu et al. [23] |
| $K_2$ | Crude oil | 0.838 | kg C/kg | Liu et al. [23] |
| $K_3$ | Natural gas | 0.590 | kg C/m$^3$ | Liu et al. [23] |
| $\delta_1$ | Human respiration | 0.079 | t C/(person·yr) | Fang [24] |
| $\delta_2$ | Pig | 0.082 | t C/(head·yr) | Kuang [25] |
| $\delta_3$ | Cattle | 0.769 | t C/(head·yr) | Kuang [25] |
| $\theta_c$ | Cultivated land | 0.374 | t/$\left(\text{hm}^2 \cdot \text{yr}\right)$ | Zhang et al. [26], Tsuruta et al. [27] |

**Table 1.** *Cont.*

| Notation | Carbon Emission Component | Coefficient | Units | Source |
|:---:|:---:|:---:|:---:|:---:|
| $\theta_1$ | Forest | −0.657 | $t/\left(hm^2{\cdot}yr\right)$ | Fang [24] |
| $\theta_2$ | Shrubland | −0.161 | $t/\left(hm^2{\cdot}yr\right)$ | Piao et al. [28] |
| $\theta_3$ | Open woodland | −0.581 | $t/\left(hm^2{\cdot}yr\right)$ | Fang [24] |
| $\theta_4$ | Other woodland | −0.103 | $t/\left(hm^2{\cdot}yr\right)$ | Piao et al. [28] |
| $\theta_5$ | High-cover grassland | −0.138 | $t/\left(hm^2{\cdot}yr\right)$ | Piao et al. [28] |
| $\theta_6$ | Medium-cover grassland | −0.046 | $t/\left(hm^2{\cdot}yr\right)$ | Piao et al. [28] |
| $\theta_7$ | Low-cover grassland | −0.021 | $t/\left(hm^2{\cdot}yr\right)$ | Fang et al. [29] |
| $\theta_8$ | Unused land | −0.005 | $t/\left(hm^2{\cdot}yr\right)$ | Lai [30] |
| $\theta_9$ | Waterbody | −0.253 | $t/\left(hm^2{\cdot}yr\right)$ | Lai [30] |

Note: The carbon sequestration coefficient per unit area of carbon sink land is expressed as a negative value.

### 2.2. Method

#### 2.2.1. Land Use Intensity

The land use intensity is calculated by the type and area of land use, and the formula is as follows [3]:

$$I = \sum (A_j \times \beta_j) / \sum A_j \tag{1}$$

where $A_j$ is the area of each land type; $\beta_j$ is the land use intensity coefficient of each land type; the land use intensity coefficients of unused land, woodland, grassland waterbody, cultivated land, and construction land take the values of 1, 2, 3, and 4, respectively [31].

#### 2.2.2. Land-Use Carbon Budget Accounting

According to previous research, the land-use carbon budget can be divided into carbon emissions from economic and social activities and carbon sequestration from natural ecosystems. Construction land and cultivated land are used as carbon source land, among which construction land hosts a series of economic and social activities, such as human production and living; here, energy consumption and human and livestock respiration are the main considerations, and pigs and cattle are considered the main livestock [13]. The carbon emission coefficient of cultivated land is $0.504\,t\,C/\left(hm^2{\cdot}\,yr\right)$ [27], while the carbon absorption coefficient is $0.130\,t\,C/\left(hm^2{\cdot}\,yr\right)$ [26], so the net carbon emission coefficient of cultivated land is $0.374\,t\,C/\left(hm^2{\cdot}\,yr\right)$. The natural ecosystem mainly uses woodland, grassland, waterbody, and unused land as carbon sink land.

(1)  Accounting for total carbon emissions

Total carbon emissions ($CE$) take into account the emissions from construction land ($C_u$) and cultivated land ($C_c$) together and is calculated as follows:

$$CE = C_u + C_c = C_e + C_p + C_c = \sum (E_m \times K_m) + \sum (p_i \times \delta_i) + A_c \times \theta_c \tag{2}$$

where $C_e$ is the carbon emission of apparent energy consumption; $E_m$ is the consumption of raw coal, crude oil, and natural gas; $K_m$ is the carbon emission coefficient of energy; $C_p$ is the carbon emission of human and livestock respiration; $p_i$ is the number of humans and livestock in a province; $\delta_i$ is the annual carbon emission per person (head); $A_c$ is the area of cultivated land; and $\theta_c$ is the carbon emission coefficient of cultivated land.

(2)  Accounting for total carbon sinks

The total carbon sink $(CS)$ of a natural ecosystem is calculated by the following formula:

$$CS = \sum CS_j = \sum (A_j \times \theta_j) \tag{3}$$

where $CS_j$ is the carbon sequestration amount of each land type; $A_j$ is the area of each carbon sink land type; and $\theta_j$ is the carbon sequestration coefficient per unit area of each carbon sink land type.

### 2.2.3. Spatial Autocorrelation Model

Spatial autocorrelation is a measure of spatial correlation that reveals the spatial interaction mechanism between research objects by describing and visualizing the spatial distribution pattern of things or phenomena to discover the degree of spatial agglomeration or dispersion. This study employs global spatial autocorrelation to reflect the overall characteristics of the degree of spatial association between carbon emissions and carbon sinks using Global Moran's I statistic with values between $-1$ and $1$ and employs local spatial autocorrelation to reveal the local aggregation characteristics of provincial units, which can be classified into four types: High-high (H-H), high-low (H-L), low-high (L-H), and low-low (L-L). The spatial autocorrelation model is a more mature approach, and the specific formulas can be found in the related literature [32,33].

### 2.2.4. Economic Contribution Coefficient of Carbon Emissions

ECC represents the economic efficiency of carbon emissions within a province and is calculated as follows [19]:

$$\text{ECC} = \frac{G_p}{G} / \frac{CE_p}{CE} \tag{4}$$

where $G_p$ and $G$ are the GDP of the $p$ provincial unit and of the whole country, respectively; $CE_p$ and $CE$ are the total carbon emissions of the $p$ provincial unit and the whole country, respectively. If ECC > 1, it indicates that the economic contribution of the provincial unit is greater than the contribution of land-use carbon emissions, which means its economic efficiency of carbon emissions is relatively high.

### 2.2.5. Ecological Support Coefficient of Carbon Emissions

ESC represents the size of the carbon sink absorption capacity within a province and is calculated as follows [19]:

$$\text{ESC} = \frac{CS_p}{CS} / \frac{CE_p}{CE} \tag{5}$$

where $CS_p$ are $CS$ are the amount of carbon sequestered in the $p$ provincial unit and in the whole country, respectively. If ESC > 1, it indicates that the carbon sequestration capacity of the province is relatively high and contributes more to the national total carbon emission reduction.

## 3. Results

### 3.1. Land Use Intensity Analysis

The overall land use intensity in China is bounded by the Heihe–Tengchong Line and was high in the southeast and low in the northwest (Figure 3). Land use intensity is influenced by a combination of many factors, including the natural landscape and the level of socioeconomic development. On the national scale, the land use intensity is 2.41, which indicates the rapid development of China's economy, accelerated industrialization and urbanization, and rapid expansion of construction land. On the regional scale, land use intensity followed the pattern Eastern region (2.66) > Central region (2.54) > Northeastern region (2.44) > Western region (2.09), and the land use intensity in the Eastern, Central, and Northeast regions is higher than the national average. On the provincial scale, land use intensity ranges from 1.46–3.01, and is higher than the national average in 13 provinces. The high values of land use intensity are concentrated in the Bohai Rim and the eastern

coastal region, with evident spatial spillover effects and linkage effects; the low values are mainly distributed in Xinjiang, Qinghai, Gansu, Inner Mongolia, and other Western regions, with low urbanization and economic development levels.

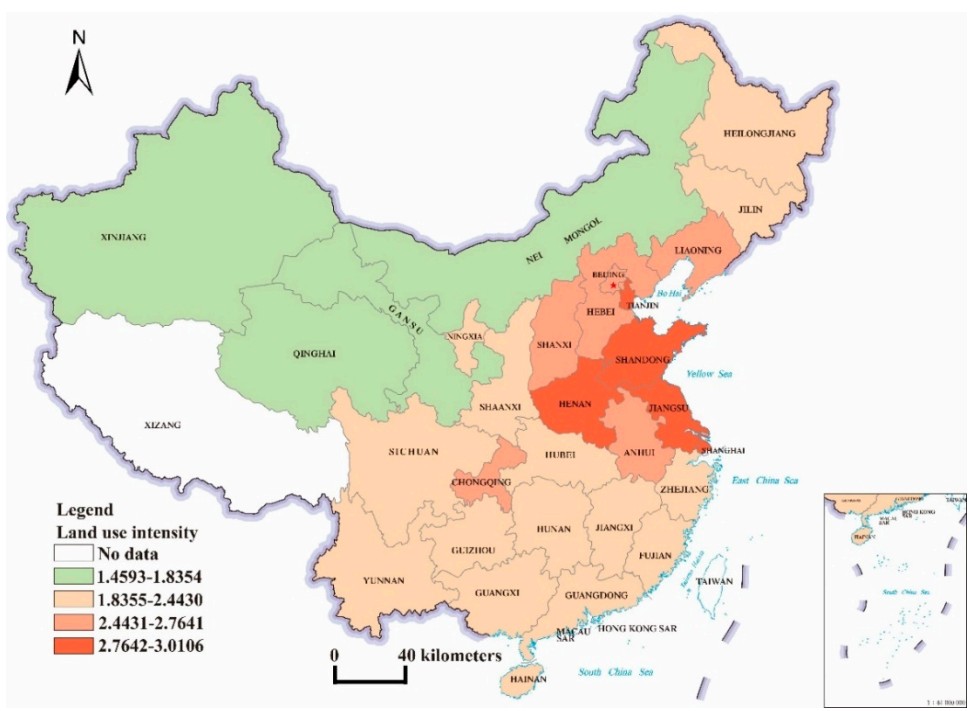

**Figure 3.** Spatial distribution of land use intensity.

### 3.2. Analysis of Carbon Source/Sink Land Area by Land Use Type

The areas of carbon sources and carbon sink land show evident spatial heterogeneity. On the national scale (Figure 4), the carbon sink land is much larger than the carbon source land, of which the carbon source area is $2.04 \times 10^8$ ha, accounting for 24.64% of the total land area, and the carbon sink land area is $6.23 \times 10^8$ ha, accounting for 75.36%. Carbon source land is mainly cultivated land, followed by construction land, and carbon sink land is mainly unused land, forest and medium-cover grassland. On the regional scale (Figure 4), the Eastern region is the only one among the four regions where the land area of the carbon source ($4.71 \times 10^7$ ha) is larger than the land area of the carbon sink ($4.57 \times 10^7$ ha). In the Central region, the land area of the carbon source is $4.73 \times 10^7$ ha and the land area of the carbon sink is $5.54 \times 10^7$ ha, and the land use type is mainly cultivated land and forest. In the Northeastern region, the area of carbon source land is $3.43 \times 10^7$ ha and the area of carbon sink land is $4.48 \times 10^7$ ha, and this region is the most concentrated area of forest resources in China at present. In the Western region, the area of carbon source land is $7.51 \times 10^7$ ha, and the area of carbon sink land is $47.72 \times 10^7$ ha. Unused land is the most dominant land type in the Western region, and only 1% of the land is used for construction. At the provincial scale (Figure 5), the area of carbon source land is larger than that of carbon sink land in eight provinces, namely, Tianjin, Hebei, Shandong, Shanghai, Jiangsu, Zhejiang, Henan, and Anhui, among which the carbon source land in Shandong and Jiangsu accounts for more than 80% of the total land area. Regarding the proportion of each land type in the total provincial land area, the distributions of cultivated land in the Eastern, Central, and Western regions are similar, showing a tortuous and fluctuating trend; in contrast, the distribution of cultivated land is relatively uniform in the Northeastern region. The provinces with a high proportion of construction land are Beijing, Tianjin, Shanghai, Shandong, Jiangsu, and Hebei; those with a high proportion of woodland are Zhejiang, Fujian, Guangdong, Hainan, Guizhou, Guangxi, Yunnan, and Northeastern provinces; grassland is mainly distributed in Inner Mongolia, Ningxia, Qinghai, Xinjiang, Shaanxi, and Gansu; unused land is mainly

distributed in Xinjiang, Gansu, Qinghai, and other Western provinces, which are located in the desert areas and permafrost zones in China.

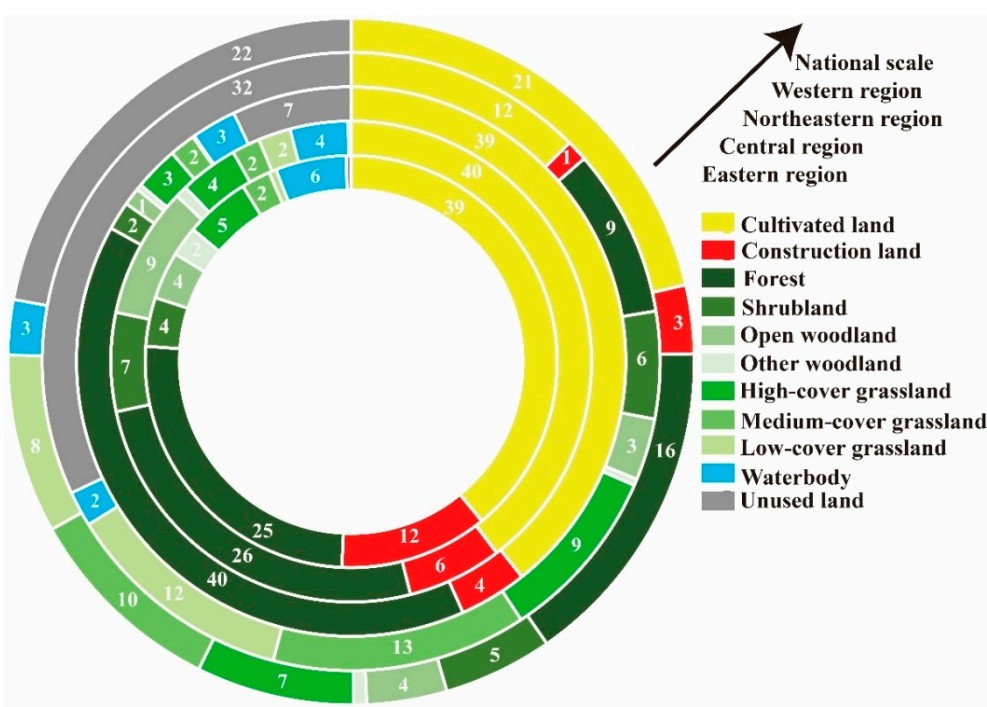

**Figure 4.** Map of land use types across the country and four regions. Note: The numbers indicate the proportion of the area of each land use type to the total land area of the region (country), and the unit is 100%, and less than 1% is not shown.

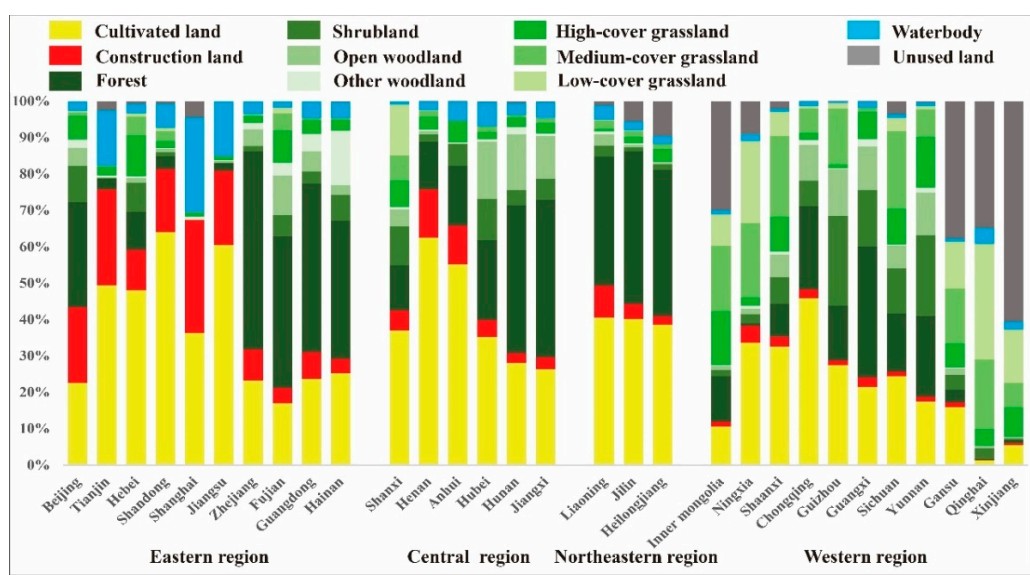

**Figure 5.** Map of provincial land use types.

### 3.3. Spatial Differentiation Characteristics of Land-Use Carbon Emissions and Carbon Sinks

3.3.1. Analysis of the Spatial Differentiation Characteristics of Land-Use Carbon Emissions

The carbon emissions of each province in China are measured based on the land use status, and the spatial distribution pattern is analyzed from three perspectives, namely, total, intensity, and per capita carbon emissions, to measure the absolute impact of total economic volume and population density on carbon emissions.

The spatial heterogeneity of total carbon emissions is evident (Figure 6a). On the national scale, the total carbon emission is $114.47 \times 10^8$ t, which is high. The carbon emissions from energy consumption are $112.37 \times 10^8$ t, accounting for 98.17%, which shows that China still needs to rely on fossil energy for its economic development. On the regional scale, in terms of the average value of total carbon emissions, the pattern follows Central region ($5.18 \times 10^8$ t) > Eastern region ($3.93 \times 10^8$ t) > Northeastern region ($3.61 \times 10^8$ t) > Western region ($3.03 \times 10^8$ t). On the provincial scale, the carbon emissions of Shanxi and Shandong are much larger than those of other provinces, with total carbon emissions of $16.42 \times 10^8$ t and $11.97 \times 10^8$ t, respectively, accounting for 14.34 and 10.45% of the total carbon emissions in China, representing a very large contribution since Shanxi and Shandong are rich in coal mining resources and have high production of industrial products such as iron, steel, and coal and are truly large fossil energy provinces. The total carbon emissions of Inner Mongolia, Jiangsu, Hebei, Shaanxi, Liaoning, Guangdong, Henan, and Xinjiang are relatively high, with an emission range of $4.76 \sim 8.56 \times 10^8$ t. Most of these provinces are around the major coal-producing provinces of Shandong and Shanxi, and their economic development relies on industry. Among them, Inner Mongolia and Liaoning have primary, heavy, and single industrial structures, and Hebei has a large proportion of the added value of secondary industries such as iron and steel; therefore, the carbon emissions of the above eight provinces are relatively high. The total carbon emissions of other provinces are relatively low for the following reasons: First, by relying on scientific and technological progress, promoting the optimization, and upgrading of the production structure and developing a circular economy, energy conservation and emission reduction are evident; second, they have a beautiful ecological environment and rich natural tourism resources and are committed to developing tourism and other tertiary industries, and their economic development is conducive to environmental protection.

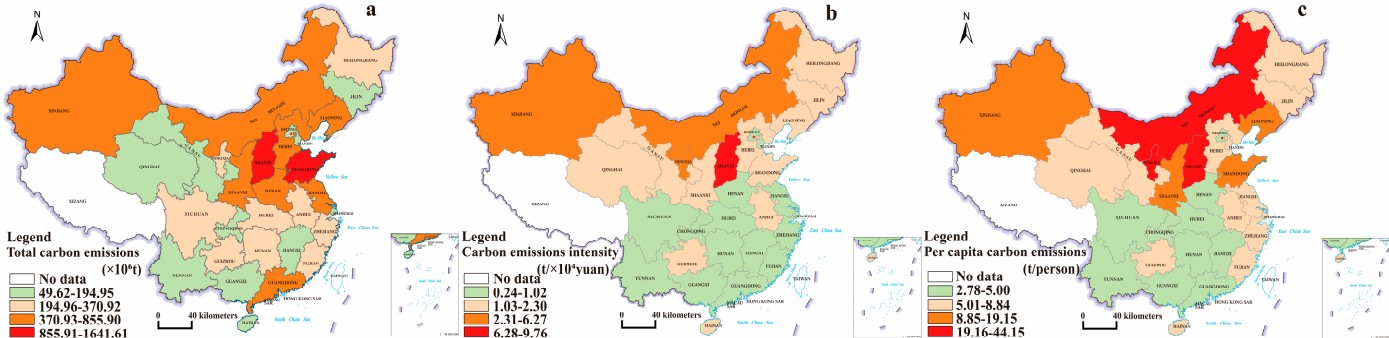

**Figure 6.** Spatial distribution of carbon emissions. Note: Subfigures (**a**–**c**) show the spatial distribution of total carbon emissions, carbon emissions intensity, and per capita carbon emissions, respectively.

The carbon emission intensity shows a spatial divergence characteristic of high in the north and low in the south, and the aggregation effect is clear, showing a scattered distribution of high values and aggregated distribution of low values (Figure 6b). On the national scale, the average carbon emission intensity is 1.80 t/ $\times 10^4$ yuan. On the regional scale, the pattern follows Western region (2.37 t/ $\times 10^4$ yuan) > Central region (2.32 t/ $\times 10^4$ yuan) > Northeastern region (1.84 t/ $\times 10^4$ yuan) > Eastern region (0.83 t/ $\times 10^4$ yuan). On the provincial scale, Shanxi dominates among high emitters, with a carbon emission intensity of 9.76 t/ $\times 10^4$ yuan, reflecting the province's low energy use efficiency and rough economic development; Xinjiang, Ningxia, and Inner Mongolia provinces are at a relatively high level of carbon emission intensity because the economic and technological development levels of these provinces are relatively backward, resulting in their carbon emission intensities being in the range of $3.90 \sim 6.27$ t/ $\times 10^4$ yuan. The percentages of provinces with low and lowest carbon emission intensities are 36.67% and 50.00%, respectively, showing an aggregated pattern of low values. The lowest values are mainly concentrated in southern China, where the carbon emission intensity is below

1.02 t/ $\times 10^4$ yuan, because the southeastern coastal provinces and Beijing–Tianjin–Hebei region are economically developed and contribute greatly to China's GDP, and the industrial structure has also changed from the traditional rough production mode to the new low-carbon-intensity development mode. With the completion of China's "eight vertical and eight horizontal" high-speed rail network system, the economic spillover effect is becoming increasingly obvious, and the rapid economic development of the Central and Western provinces promotes the optimization and upgrading of industrial structure and improving energy efficiency.

The per capita carbon emissions show a spatial divergence characteristic of "high north and low south", with a more apparent bifurcation (Figure 6c). On the national scale, the average is 10.02 t/person. On the regional scale, carbon emissions in the Western region (12.49 t/person) > Central region (10.93 t/person) > Northeastern region (9.58 t/person) > Eastern region (6.89 t/person). On the provincial scale, the per capita carbon emissions of Shanxi are as high as 44.15 t/person, 15.9 times that of the lowest carbon emissions, observed in Sichuan province (2.78 t/person), and only seven provinces are above the national average, namely, Inner Mongolia and Xinjiang because of their small population density, and Shanxi, Shandong, Shaanxi, Liaoning, and Ningxia because they are high-carbon-emission provinces, resulting in their per capita carbon emissions is at the higher level. The coastal and southern provinces have a high population density, a high level of economic development, and a relatively green production structure; therefore, the per capita carbon emissions are at the lower and lowest level.

### 3.3.2. Analysis of the Spatial Differentiation Characteristics of Land-Use Carbon Sinks

The carbon sinks of different land types vary at high and low levels, with large differences. Among them, the carbon sink of the forest was 85.33 $\times 10^6$ t, followed by open woodland and high-cover grassland, with carbon sinks of 18.13 $\times 10^6$ t and 8.48 $\times 10^6$ t, respectively. The next most abundant carbon sinks were in shrubland, waterbody, medium-cover grassland, and low-cover grassland, with values of 6.86 $\times 10^6$ t, 5.42 $\times 10^6$ t, 3.63 $\times 10^6$ t, and 1.46 $\times 10^6$ t, respectively. The lowest carbon sinks were in unused land and other woodland, with only 9.14 $\times 10^5$ t and 5.28 $\times 10^5$ t, respectively.

The total land-use carbon sink shows the spatial characteristic of "high in the north-south and low in the middle" (Figure 7a). On the national scale, the total carbon sink is 130.76 $\times 10^6$ t. According to the land use type map in Figure 4, the three types of forest, open woodland, and high-cover grassland only account for 27% of the national land area, but the carbon sink contribution value accounts for 85.61%. On the regional scale, to eliminate the effect of the number of provinces, the average value of total carbon sinks is taken, with the Northeastern t region (7.42 $\times 10^6$ t) > Western region (5.70 $\times 10^6$ t) > Central region (4.24 $\times 10^6$ t) > Eastern region (2.04 $\times 10^6$ t). On the provincial scale, the total carbon sinks of Inner Mongolia, Heilongjiang, and Yunnan all exceed 10 $\times 10^6$ t, which is the highest level. These provinces are rich in forest and grass resources, which have become important carbon sink resource provinces and ecological security barriers in China. The provinces with high total carbon sinks are Sichuan, Guangxi, Hunan, Guangdong, Jiangxi, Xinjiang, Jilin, and Hubei, with total carbon sinks ranging from 5.11 to 9.10 $\times 10^6$ t, and these provinces are richer in forest resources. The total amount of carbon sinks in most provinces in China is below 5.00 $\times 10^6$ t, and the function of carbon sinks is weak. In the process of urbanization, the construction land is expanding rapidly, which constricts the living space of carbon sink land. The spatial distribution characteristics of the total carbon sinks match the natural resource endowment of China.

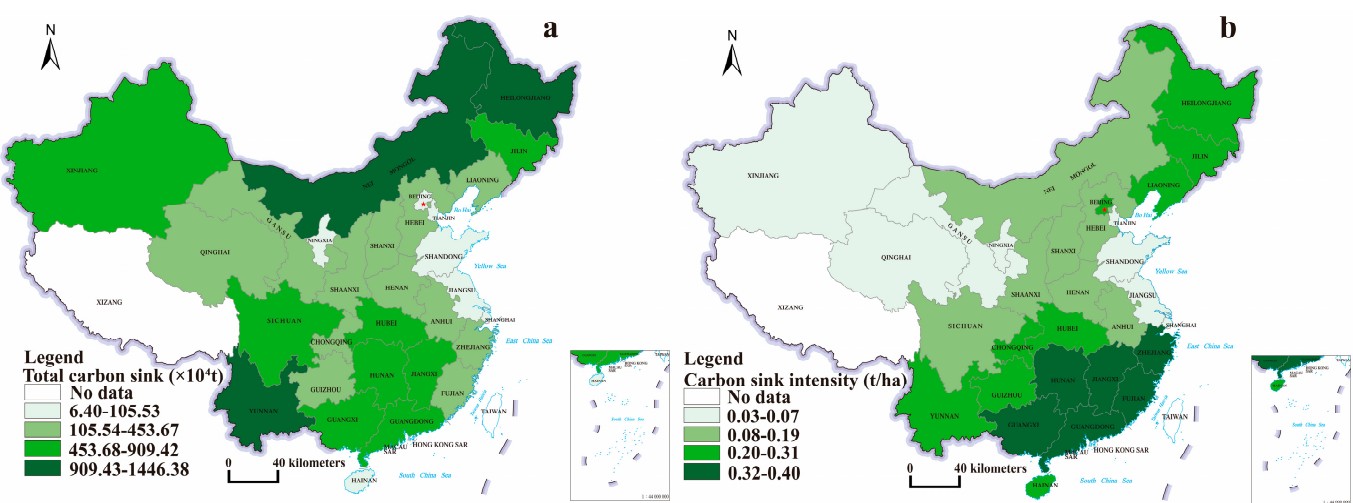

**Figure 7.** Spatial distribution of carbon sinks. Note: Subfigures (**a**,**b**) show the spatial distribution of total carbon sink and carbon sink intensity, respectively.

To eliminate the absolute influence of provincial area on the total carbon sink, this study applies carbon sink intensity to measure the $CO_2$ absorption capacity per unit area. The overall carbon sink intensity shows a characteristic of continuous decreasing intensity from southeast to northwest (Figure 7b). On the national scale, the average value of carbon sink intensity is 0.20 t/ha. On the regional scale, the Northeastern region (0.28 t/ha) > Central region (0.24 t/ha) > Eastern region (0.20 t/ha) > Western region (0.16 t/ha). On the provincial scale, the highest carbon sink intensity is located in six provinces, namely, Zhejiang, Jiangxi, Hunan, Fujian, Guangdong, and Guangxi, all of which have an intensity greater than 0.32 t/ha, followed by Yunnan, Guizhou, Chongqing, Hubei, and the three Northeastern provinces, followed by seven provinces, namely, Sichuan, Shaanxi, Henan, Anhui, Shanxi, Hebei, and Inner Mongolia, where the carbon sink intensity is between 0.1 and 0.2 t/ha; the rest of the provinces have a carbon sink intensity of 0.1 t/ha or less. According to Figure 5, it can be seen that the carbon source land in Tianjin, Shandong, Jiangsu, and Shanghai is much larger than the carbon sink land, while the woodland in Xinjiang, Qinghai, and Gansu accounts for a very small proportion and unused land accounts for a great proportion, but their carbon sink function is not significant, thus leading to low carbon sink efficiency in these provinces.

### 3.3.3. Spatial Autocorrelation Analysis

According to Tobler's First Law, spatial adjacency factors influence the spatial distribution of carbon emissions and carbon sinks to a certain extent. The global spatial autocorrelation results show that the global Moran's I of total carbon emissions, carbon emission intensity, per capita carbon emissions, total carbon sink, and carbon sink intensity are 0.191, 0.268, 0.277, 0.286, and 0.559, respectively, indicating a significant positive spatial correlation and spatial aggregation characteristics, among which the positive correlation effect of carbon sink intensity is the strongest.

To further study the local spatial correlation between carbon emissions and carbon sinks, Local Indicators of Spatial Association (LISA) spatial distribution maps were drawn using GeoDa and ArcGIS (Figures 8 and 9). It can be found that carbon emissions and carbon sinks mainly show H-H clusters and L-L clusters, in which the local aggregation effects of carbon emission intensity, per capita carbon emissions, and carbon sink intensity are the most significant. In terms of total carbon emissions (Figure 8a), H-H types were distributed in Inner Mongolia, Shanxi, Henan, and Hebei, where total carbon emissions were high near the region; L-L types were distributed in Sichuan, Guizhou, and Hunan. In terms of carbon emission intensity (Figure 8b), L-L types are mainly distributed in southeastern coastal and Central provinces, and the occurrence of H-L and L-H types

indicates the obvious heterogeneity of these provinces. In terms of per capita carbon emissions (Figure 8c), H-H types were distributed in five provinces, Inner Mongolia, Ningxia, Shaanxi, Shanxi, and Liaoning; L-L types were mainly distributed in southern China, and no H-L types were present. In terms of the total carbon sink (Figure 9a), H-H clusters are distributed in provinces with abundant forest resources, L-H types were distributed in Guizhou and Liaoning, and no H-L types existed. In terms of carbon sink intensity (Figure 9b), H-H clusters were distributed in six provinces in southern China and L-L clusters were distributed in provinces north of the Qinling Mountains-Huaihe River Line, with obvious local spatial correlation effects.

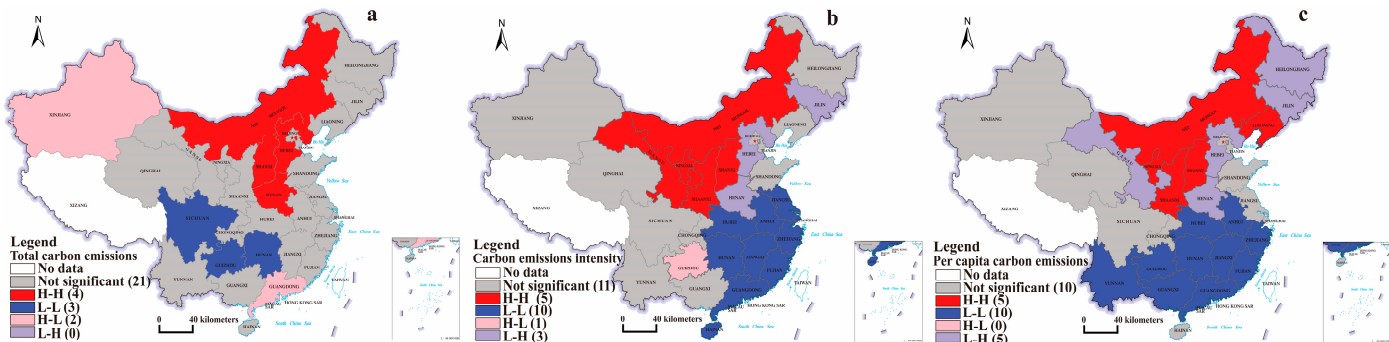

**Figure 8.** The LISA spatial agglomeration pattern of carbon emissions. Note: Subfigures (**a**–**c**) show the LISA spatial agglomeration patterns of total carbon emissions, carbon emissions intensity, and per capita carbon emissions, respectively.

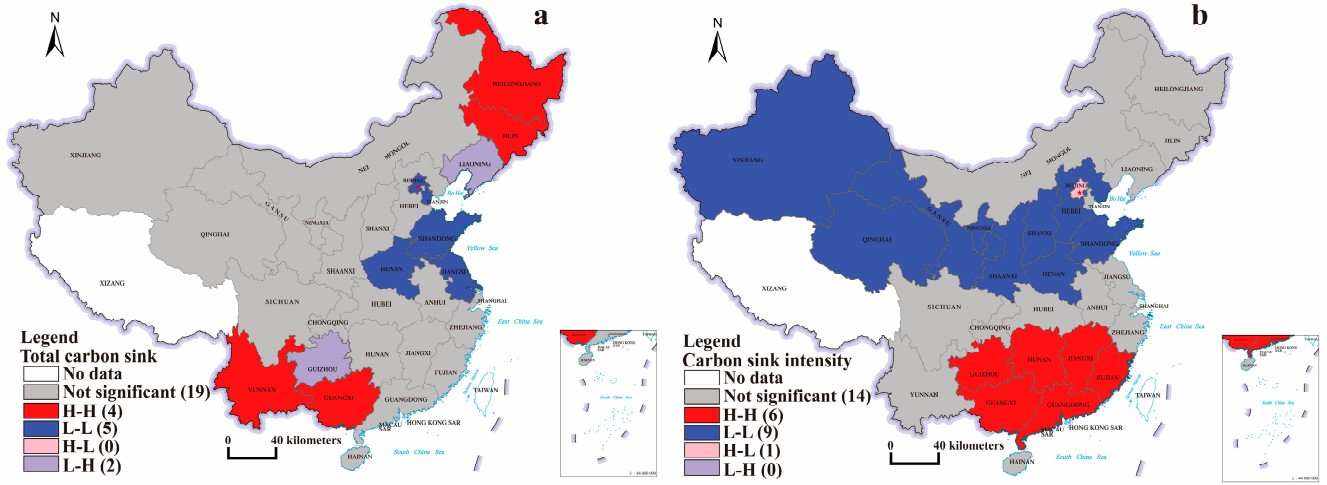

**Figure 9.** The LISA spatial agglomeration pattern of carbon sinks. Note: Subfigures (**a**,**b**) show the LISA spatial agglomeration patterns of total carbon sink and carbon sink intensity, respectively.

*3.4. Carbon Balance Zoning and Low-Carbon Development Proposal*

3.4.1. Analysis of the Spatial Characteristics of ECC and ESC

Land use intensity can reflect both the level of economic development and be an effective regulatory tool for industrial development. The low ECC values show aggregation characteristics and are mainly distributed in northwest China (Figure 10a). The provinces of Xinjiang, Qinghai, Gansu, and Inner Mongolia, where land use intensity and ECC are highly coupled, all show the lowest values; Shanxi, Shaanxi, Heilongjiang, and Liaoning, where the production mode is still dominated by industry and the economic development mode is rough, have ECC values less than 0.72. Thanks to the rapid economic development in recent years, 17 provinces in China have ECC values of over 1.00, mainly in the Eastern and Central regions of the country, where the economic efficiency is higher, and the distribution characteristics of carbon emission intensity are similar. Beijing is the political, economic,

and cultural center of China, with an ECC of 5.25, 40 times higher than that of the province with the lowest value, Shanxi (0.13), with a clear bifurcation.

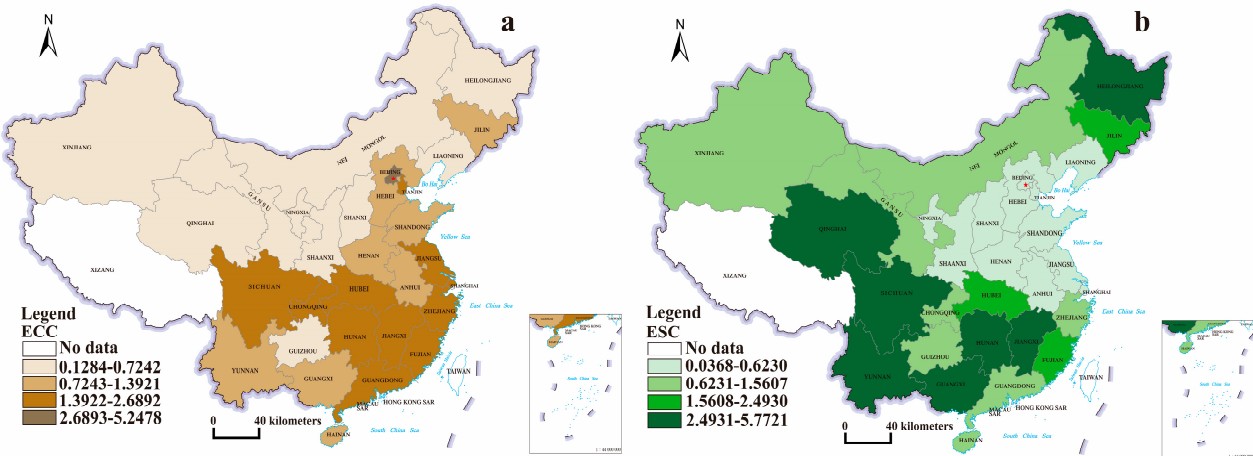

**Figure 10.** Spatial distribution of ECC and ESC. Note: Subfigures (**a**,**b**) show the spatial distribution of ECC and ESC, respectively.

With the gradual deepening of the concept of "lucid waters and lush mountains are invaluable assets", China's ecological civilization construction is accelerating, the national average level of ESC is 1.58, and there are 17 provinces that exceed 1.00, so the carbon sink capacity is more significant. According to the land use intensity map (Figure 3) and the ESC spatial distribution map (Figure 10b), the land use pattern can deeply influence the ecological carrying capacity, and the two show highly coupled characteristics. With the Beijing–Tianjin–Hebei region as the center and spreading outward to Liaoning, Hebei, Shandong, and other provinces with the highest land use intensity, rapid urbanization, and an expanding proportion of carbon source land, the ESC of the region was at the lowest level, with all values less than 0.63. The land use intensity and total carbon emissions in the area south of the Qinling Mountains-Huaihe River Line are at a low level, so the ESC of the area is better.

### 3.4.2. Carbon Balance Zoning

According to existing research [19,34], an ECC greater than 1 indicates that the economic contribution of a province is greater than its carbon emissions contribution, while an ECC less than 1 indicates that a province is generally less productive and its high $CO_2$ emissions will harm other regions. An ESC greater than 1 means that the province's share of carbon absorption is greater than its carbon emissions and that it has a high carbon ecological capacity, which can bring positive externality effects to other regions, while an ESC less than 1 indicates that the province's carbon emissions need to be shared by other regions. In this study, according to the coupling of ECC and ESC, 30 provinces in China were divided into four zones: A low-carbon development zone, a carbon sink function zone, a carbon intensity control zone, and a high-carbon optimization zone (Table 2), and it can be seen that the zones are spatially aggregated (Figure 11).

**Table 2.** Carbon balance zoning based on ECC and ESC.

| Zone Name | Conditions | Province |
|---|---|---|
| Low-carbon development zone | ECC >1 & ESC >1 | Sichuan, Chongqing, Hubei, Hunan, Jiangxi, Fujian, Guangdong, Guangxi, Yunnan, Hainan |
| Carbon sink function zone | ECC < 1 & ESC > 1 | Xinjiang, Gansu, Qinghai, Inner Mongolia, Jilin, Heilongjiang, Guizhou |
| Carbon intensity control zone | ECC > 1 & ESC < 1 | Beijing, Tianjin, Henan, Anhui, Jiangsu, Shanghai, Zhejiang |
| High-carbon optimization zone | ECC < 1 & ESC < 1 | Ningxia, Shaanxi, Shanxi, Hebei, Shandong, Liaoning |

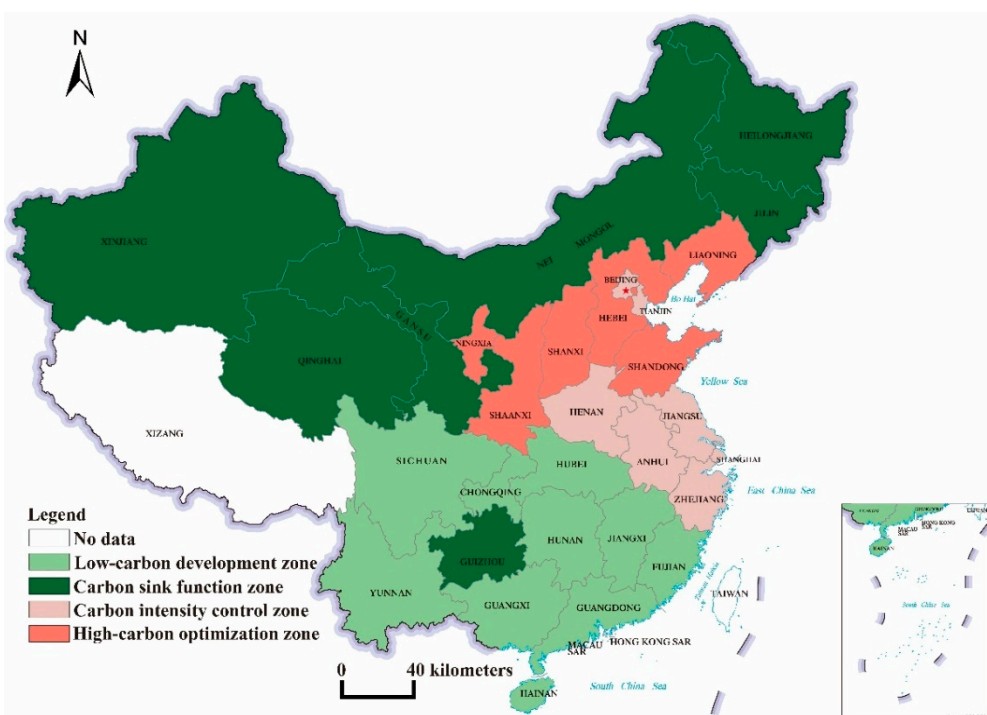

**Figure 11.** Spatial distribution of carbon balance zones.

The low-carbon development zone is clustered and distributed to the south of the Qinling Mountains-Huaihe River Line, containing 10 provinces, and the ECC and ESC of this area are both high. The reason is that the region has low land use intensity, a well-developed water system, and moist and fertile soil, which is conducive to the development of paddy field agriculture, with agricultural production areas such as South China and the Yangtze River basin. The plant types are mostly evergreen broad-leaved forests, and there are many ecological function areas, such as Sichuan and Yunnan forests, the central Hainan Island tropical rainforest, Nanling mountains, etc. The carbon sink function is more significant, reaching $61.37 \times 10^6$ t, and with the Beibu Gulf, Pearl River Delta, the middle reaches of the Yangtze River, the cross-strait economic zone, and other urban clusters and national key development areas, there is a strong economic base and scientific and technological innovation capabilities.

The carbon sink function zone is mainly located in the northwest and northeast regions, including seven provinces, whose land use intensity is at the lowest level nationwide, with a beautiful natural environment and a total carbon sink of $48.15 \times 10^6$ t. Although its economic development level is low, the ecological environment is better, and the carbon sink function is strong. Its northern sand control belt and northeastern forest belt contain several ecological functional areas, such as the DaXingAnLing forests and the Hulunbeir Grassland, whose ecological security strategy is of great significance.

The carbon intensity control zone is mainly distributed in the Yangtze River Delta and Beijing–Tianjin region, containing seven provinces, whose land use intensity is drastic, with a total carbon emission of $2.18 \times 10^9$ t and carbon sink of $9.07 \times 10^6$ t, and a more severe carbon deficit. The Yangtze River Delta city cluster is one of the six internationally recognized world-class city clusters, an important international gateway to the Asia-Pacific region, and the number one economic zone in China. Beijing and Tianjin drive the development of the Beijing–Tianjin–Hebei city cluster and the Bohai Rim. The carbon intensity control zone has a high level of economic development, relying mainly on manufacturing and high-tech industries, and its economic contribution is large, showing positive externalities and playing an important role in China's economic development. Moreover, the region has a high level of urbanization, and the increase in population density leads to the expansion of construction land and the reduction of carbon sink land, and the carbon sink is greatly reduced.

The high-carbon optimization zone is mainly distributed in the Bohai Rim region, containing six provinces, whose land use level is strong. It is an important energy raw material and coal chemical and manufacturing base in China, which is significant in meeting China's energy demand and export, but energy extraction needs to rely on industry, and economic development is mainly rough. Therefore, its total carbon emissions are $4.77 \times 10^9$ t, and the carbon sink is only $12.16 \times 10^6$ t. The carbon deficit is the most severe.

### 3.4.3. Low-Carbon Development Proposal

According to the natural background properties, economic development level, ecological environment, and resource structure of each zone, we propose differentiated low-carbon development proposals in terms of ecological environmental protection and low-carbon economic development according to local conditions.

In terms of ecological environmental protection, the low-carbon development zone should continue to strengthen the protection of nature reserves, important wetlands, mudflats, and water reserves; promote the ecological environment, basic farmland, and other protection planning and intensive land use; reduce the impact of industrialization and urbanization on the ecological environment; avoid problems such as excessive land occupation, overexploitation of water resources, and excessive pressure on the ecological environment; and strive to improve environmental quality. The carbon sink function zone should continue to promote the protection of natural forest resources, the return of grazing to grass, and implement wind and sand source management and the construction of protective forest system tracts; increase the carbon sequestration capacity of terrestrial ecosystems, take advantage of local conditions to actively develop wind energy, solar energy, geothermal energy, and other clean energy; and continue to play the ecosystem function and ecological barrier roles. The carbon intensity control zones should optimize the layout of cities and towns; control urban sprawl and expansion, industrial blooming, and overdispersed development zones; strictly control development intensity; increase investment in ecological environmental protection; strengthen environmental treatment and ecological restoration; effectively and strictly protect natural resources such as cultivated land, water, wetlands, woodlands, and grasslands; protect the green open space between cities; and build ecological corridors. The high-carbon optimization zone should make ecological restoration and environmental protection a binding goal that must be achieved, enhance the responsibility of enterprises for environmental protection, implement comprehensive environmental pollution improvement projects, strengthen ecological restoration in industrial pollution areas, and build an ecological pattern consisting of mountains, urban green areas, and regional ecological water networks.

In terms of low-carbon economic development, the low-carbon development zone should continuously improve the capacity for independent innovation, gather innovative elements, enhance industrial agglomeration capacity, ensure the quality and efficiency of economic development, and form a modern industrial system of division of labor and collaboration; at the same time, it should continue to rely on the advantages of location, grow the comprehensive strength of the city, improve the living environment, enhance the ability to gather the population, accelerate the opening of the border areas to the outside world, and form a new window for the opening up of China and strategic space. The carbon sink function zone should rely on the advantages of local resources to develop the processing industry of agricultural and animal husbandry products, cultivate special agricultural and animal husbandry industries, develop intensive, standardized, and efficient breeding, and promote the transformation of agricultural development; at the same time, it should strengthen the exploration and development of oil and natural gas in the northeast region and Xinjiang, and accelerate the construction of energy transmission channels, but adhere to the principle of "development at the point and protection on the surface" to promote economic development, improve people's living standards, and lay the foundation for ecological protection through development at the point, while achieving protection on the surface. The carbon intensity control zone should increase the development and applica-

tion of technology for the conservation and efficient use of energy resources, accelerate technological innovation and upgrading, take the lead in improving the efficiency of energy use, and play a role as a model for the national carbon emission intensity reduction; at the same time, it should accelerate the development of modern service industries, high-tech industries, and high value-added agriculture, optimize the layout of production space, living space, and ecological space, promote the functional complementation and economic connection between cities, and enhance the economic development of the surrounding areas. The high-carbon optimization zone should foster the rational development of coal and other energy resources, the use of advanced production technology to promote the optimization and upgrading of mining structure strengthen comprehensive utilization, and continue to play the function of guaranteeing national energy security; at the same time, it should determine the carbon emission intensity target and total carbon emission constraints, guide the development of industrial clusters, adjust and constrain the development of high-carbon industries, promote the transformation of resource-based cities, promote regional low-carbon transformation, and strive to create a resource-based economic transformation demonstration area.

## 4. Discussion

In terms of land use pattern and intensity, China exhibits strong land use intensity, with high values concentrated in the Bohai Rim and the eastern coastal region and low values mainly distributed in the Western region, which is similar to the findings of Chen et al. [35]. The area of carbon sink land in China is much larger than that of carbon source land, and the land use types are mainly grassland, woodland, cultivated land, and unused land [36]. Among them, carbon source land is mainly cultivated land, followed by construction land, which is in line with the reality of a largely agricultural country and the continuous expansion of construction land in China [37,38], while carbon sink land is mainly woodland, grassland and unused land.

In recent years, with the rapid development of China's economy, total carbon emissions have continued to increase and have been confirmed by the academic community [39,40]. Jing et al. found that the total carbon emissions in China were $109.5 \times 10^8$ t in 2010 and $138.5 \times 10^8$ t in 2019, with an average annual growth rate of 2.37% [41]. In this study, the total carbon emissions in China in 2018 were $114.47 \times 10^8$ t, which is still considered high, and this value is still in a reasonable range compared with previous studies and is more accurate [39,41], which considers the apparent energy consumption (raw coal, crude oil, and natural gas) and use a carbon emission coefficient that is more in line with China's actual situation. Construction land is the main source of carbon emissions in China and cultivated land accounts for only a very small portion, which is consistent with previous research results [41,42]. The aggregation effect of carbon emission intensity in China is obvious, and in general, it shows a scattered distribution of high values, an aggregated distribution of low values, and a decreasing trend from west to east and from north to south [43]. The total national carbon sink is $130.76 \times 10^6$ t, and woodlands and grasslands account for the majority of the carbon sink; woodlands and grasslands are the main carriers of terrestrial carbon absorption in China [44,45] and have the most obvious offsetting effect on carbon emissions.

China is a large carbon-emitting country; hence, it is more important to reduce carbon emission sources than to increase carbon sinks [46]. China has a large population and limited land resources, and in the rapid urbanization, the expansion of construction land is bound to squeeze other land use types, which leads to a rapid growth trend of carbon emissions [47]. Therefore, limiting the occupation of carbon sink land by construction land is significant for carbon emission reduction in China. In terms of the industrial system, the development of green industries such as the service industry and high-tech industries has become the first choice [48]. The carbon intensity control zone and high-carbon optimization zone have significant responsibilities for reducing carbon emissions, and they should accelerate technological innovation and upgrades [49] and optimize the industrial

structure [13]. These measures can improve energy use efficiency and promote the transformation of resource-based cities, which will help achieve low-carbon development. To achieve carbon neutrality in China, it is very difficult to rely solely on the increase in carbon sequestration. However, this does not mean that ecological protection and afforestation should be neglected. A good natural environment is not only the largest contributor to carbon sinks [50], but also provides a variety of ecosystem service functions such as soil and water conservation and conservation of species diversity [51]. Therefore, the low-carbon development zone and carbon sink function zone should focus on protecting ecological resources, greatly improving carbon absorption [36], and avoiding overexploitation; most of the national ecological protection projects are distributed within these two areas [52]. The carbon intensity control zone and high-carbon optimization zone should increase investment in ecological environmental protection, strengthen environmental management and ecological restoration [53], optimize the layout of towns and cities, intensify land use, and build an ecological pattern of urban green areas to provide more ecological functions [54].

## 5. Conclusions

Based on the land use perspective, this work analyzes the land use intensity and the distribution of carbon source/sink land in China, determines the carbon budget of each province, combines ECC and ESC for carbon balance zoning, and proposes differentiated low-carbon development suggestions. The findings of the study are as follows:

(1) The intensity of land use in China is strong, divided by the Heihe–Tengchong Line, showing the characteristic of "high in the southeast and low in the northwest", with high values concentrated in the Bohai Rim and the eastern coastal region and low values mainly distributed in the Western region.

(2) The spatial heterogeneity of China's total carbon emissions is apparent, and the emission state is still considered high. Both carbon emission intensity and carbon emissions per capita show the spatial heterogeneity characteristics of "high in the north and low in the south".

(3) The total carbon sink in China shows the spatial characteristic of "high in the north-south and low in the middle", which matches the natural resource endowment of China. The overall carbon sink intensity shows a characteristic of continuous decreasing intensity from southeast to northwest.

(4) Total carbon emissions, carbon emission intensity, per capita carbon emissions, total carbon sinks, and carbon sink intensity all have positive global spatial correlations and significant local aggregation effects.

(5) Carbon balance zoning shows a clear aggregation effect. The low-carbon development zone is distributed south of the Qinling Mountains–Huaihe River Line, the carbon sink function zone is mainly distributed in the northwest and northeast, the carbon intensity control zone is mainly distributed in the Yangtze River Delta and Beijing–Tianjin area, and the high-carbon optimization zone is mainly distributed in the Bohai Rim. Development proposals are put forward for both ecological environmental protection and low-carbon economic development.

Due to data source problems, this study is only relevant for 2018, and no analysis in terms of spatial and temporal evolution was carried out. However, this study revisits China's terrestrial carbon balance, which is crucial for understanding the carbon cycle and sustainable development, and the results can provide a reference for the formulation of China's territorial spatial planning.

**Author Contributions:** Writing—original draft preparation, H.W. and F.W.; writing—review and editing, H.W. and F.W.; visualization, Y.L.; data collection and analysis, Z.L. and X.C.; funding acquisition, F.W. All authors have read and agreed to the published version of the manuscript.

**Funding:** This research is supported by the National Key R&D Program of China (2019YFD0901301), the National Natural Science Foundation of China (No. 42261064), and the Hainan Federation of Social Sciences (HNSK (YB)19-09).

**Institutional Review Board Statement:** Not applicable.

**Informed Consent Statement:** Not applicable.

**Data Availability Statement:** Publicly available datasets were analyzed in this study. These data can be found here: https://www.resdc.cn (accessed on 30 August 2022), https://www.ceads.net.cn/ (accessed on 30 August 2022), and *China statistical yearbook*. http://www.stats.gov.cn/tjsj/ndsj/ (accessed on 30 August 2022) [22].

**Conflicts of Interest:** The authors declare that they have no conflict of interest.

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
