# Peer review of "Spatial Differentiation of Carbon Budgets and Carbon Balance Zoning in China Based on the Land Use Perspective"

_sustainability, doi:10.3390/su142012962_

Round 1

Reviewer 1 Report

This paper reports a thorough investigation into the spatial characteristics of carbon emissions and carbon sinks in 30 Chinese provinces. The paper provides some useful findings and observations, so it is recommended for publication in Sustainability, although some comments should be addressed as listed below.

1. Fig. 6: Why is the carbon sink value of Jiangsu province so low?

2. More in-depth discussions should be added to further explain the statements made in Section 3 based on the data.  

3. It would be better if the data on Taiwan can be included in the paper.

4. What do the numbers in Figure 3 stand for?

Author Response

Response to Reviewer 1 Comments

Dear Reviewer:

We highly appreciate the reviewer for the insightful comments and criticism, which have helped us improve both the content and the presentation of our work. We believe that the revised version of our paper addresses all concerns by the referees in detail. The changes are highlighted within the manuscript. Please see below, in red, for a point-by-point response to the reviewer’s comments and concerns.

Point 1: Fig. 6: Why is the carbon sink value of Jiangsu province so low?

Response 1: We sincerely appreciate the reviewer’s question. Combined with Figure 5. Map of provincial land use types, it can be seen that the land use types in Jiangsu Province are mainly cultivated land, construction land and waterbody, while the area of woodlands and grasslands is 4,213 km2, accounting for only 4.06%, so the total carbon sink value is 58.33×104 t.

Point 2: More in-depth discussions should be added to further explain the statements made in Section 3 based on the data.

Response 2: Thanks for your suggestion. we have added a discussion section according to your advice. First, we have added literature citations. Second, we have fully discussed the findings of this study in terms of land use intensity, carbon emissions and carbon sinks, as follows:

In terms of land use pattern and intensity, China exhibits strong land use intensity, with high values concentrated in the Bohai Rim and the eastern coastal region and low values mainly distributed in the western region. which is similar to the findings of Chen et al. [35]. The area of carbon sink land in China is much larger than that of carbon source land, and the land use types are mainly grassland, woodland, cultivated land, and unused land [36]. Among them, carbon source land is mainly cultivated land, followed by construction land, which is in line with the reality of a large agricultural country and the continuous expansion of construction land in China [37,38], while carbon sink land is mainly woodland, grassland and unused land.

In recent years, with the rapid development of China's economy, total carbon emissions have continued to increase and have been confirmed by the academic community [39,40]. Jing et al found that the total carbon emissions in China were 109.5×108 t in 2010 and 138.5×108 t in 2019, with an average annual growth rate of 2.37% [41]. In this study, the total carbon emissions in China in 2018 were 114.47×108 t, which is still considered high, and this value is still in a reasonable range compared with previous studies and is more accurate than the findings of previous studies, which consider the apparent energy consumption (raw coal, crude oil and natural gas) and use a carbon emission coefficient that is more in line with China's actual situation. Construction land is the main source of carbon emissions in China and cultivated land accounts for only a very small portion, which is consistent with previous research results [41,42]. The aggregation effect of carbon emission intensity in China is obvious, and in general, it shows a scattered distribution of high values, an aggregated distribution of low values and a decreasing trend from west to the east and from north to south [43]. The total national carbon sink is 130.76×106 t, and woodlands and grasslands account for the majority of the carbon sink; woodlands and grasslands are the main carriers of terrestrial carbon absorption in China [44,45] and have the most obvious offsetting effect on carbon emissions.

Some of the references added are as follows:

  1. Chen, W.X.; Zeng, J. Decoupling analysis of land use intensity and ecosystem services intensity in China. Nat. Resour. 2021, 36, 2853-2864.
  2. Li, J.S.; Guo, X.M.; Chuai, X.W.; Xie, F.J.; Yang, F. Reexamine China’s terrestrial ecosystem carbon balance under land use-type and climate change. Land Use Policy. 2021, 102, 105275.
  3. Tang, X.L.; Lu, C.Y.; Meng, P.; Cheng, W. Spatiotemporal Evolution of the Environmental Adaptability Efficiency of the Agricultural System in China. Sustainability. 2022, 14, 3685.
  4. Zhu, M.C.; Shen, L.Y.; Tam, V.W.Y.; Liu, Z.; Shu, T.H.; Luo, W.Z. A load-carrier perspective examination on the change of ecological environment carrying capacity during urbanization process in China. Total Environ. 2020, 714, 136843.
  5. Xu, W.H.; Xie, Y.L.; Ji, L.; Cai, Y.P.; Yang, Z.F.; Xia, D.H. Spatial-temporal evolution and driving forces of provincial carbon footprints in China: An integrated EE-MRIO and WA-SDA approach. Eng. 2022, 176, 106543.
  6. Lin, Q.W.; Zhang, L.; Qiu, B.K.; Zhao, Y.; Wei, C. Spatiotemporal Analysis of Land Use Patterns on Carbon Emissions in China. Land. 2021, 10, 141.
  7. Jing, X.D.; Tian, G.L.; Li, M.R.; Javeed, S.A. Research on the Spatial and Temporal Differences of China’s Provincial Carbon Emissions and Ecological Compensation Based on Land Carbon Budget Accounting. J. Environ. Res. Pub He. 2021, 18, 12892.
  8. Huang, H.Q.; Zhou, J. Study on the Spatial and Temporal Differentiation Pattern of Carbon Emission and Carbon Compensation in China’s Provincial Areas. Sustainability. 2022, 14, 7627.
  9. Yang, M.; Liu, Y.S.; Tian, J.Z.; Cheng, F.Y.; Song, P.B. Dynamic Evolution and Regional Disparity in Carbon Emission Intensity in China. Sustainability. 2022, 14, 4052.
  10. Wang, Y.L.; Wang, X.H.; Wang, K.; Chevallier, F.; Zhu, D.; Lian, J.H.; He, Y.; Tian, H.Q.; Li, J.S.; Zhu, J.X.; Jeong, S.J.; Canadell, J.G. The size of the land carbon sink in China. Nature. 2022, 603, E7-E9.
  11. Wang, J.; Feng, L.; Palmer, P.I.; Liu, Y.; Fang, S.X.; Bösch, H.; O'Dell, C.W.; Tang, X.P.; Yang, D.G.; Liu, L.X.; Xia, C.Z. Large Chinese land carbon sink estimated from atmospheric carbon dioxide data. Nature. 2020, 586, 720-723.

Point 3: It would be better if the data on Taiwan can be included in the paper.

Response 3: Thank you for your valuable comments. In order to increase the comprehensiveness and comparability of the paper, we initially wanted to study the carbon budget for all provinces in China, and for this purpose we tried to obtain data from many sources. Unfortunately, there is a lack of data on Taiwan, so only 30 provincial-level administrative regions are studied in this paper.

Point 4. What do the numbers in Figure 3 stand for?

Response 4: Thanks for your careful check. The meaning of the numbers has been explained in the paper. The numbers indicate the proportion of the area of each land use type to the total land area of the region (country).

We look forward to hearing from you regarding our submission. We would be glad to respond to any further questions and comments that you may have.

Sincerely,

Authors

23th September, 2022

Reviewer 2 Report

Dear authors,

Please find in the attached file my comments to this manuscript.

Thank you.

Author Response

Response to Reviewer 2 Comments

Dear Reviewer:

We highly appreciate the reviewer for the insightful comments and criticism, which have helped us improve both the content and the presentation of our work. We believe that the revised version of our paper addresses all concerns by the referees in detail. The changes are highlighted within the manuscript. Please see below, in red, for a point-by-point response to the reviewer’s comments and concerns.

Point 1: The novelty/new contribution of this study regarding the existing literature is not demonstrated. What this study adds to the literature on this topic? The last two paragraphs of the Introduction tried to demonstrate this new contribution (studies on the national scale are rare, utilization of land use subdivisions to map carbon emissions and sinks), but there are already studies with similar approaches, such as Wang et al. (2021). Try to be more convincing and clear about the new contribution of this study.

Response 1: We sincerely appreciate the reviewer’s question. We have reorganized the logic of the introduction part and further condensed and summarized it to highlight the research contribution and innovative value of this article, as follows:

In general, there have been systematic studies on carbon emissions, carbon sinks and the land use carbon balance, which are important references for this work; however, the following problems are still worth exploring in depth. First, the subdivision of land use types is not deep enough, which leads to large errors in the measurement results of the land use carbon budget. Second, most of the carbon emission factors for energy consumption use data published by the IPCC; however, China, as a large energy-consuming country, needs to determine its carbon emission factors through field measurements. Third, most scholars analyze the land use carbon budget from two perspectives, total carbon emissions and total carbon sinks, but fail to study it from multiple perspectives, such as carbon emission intensity, per capita carbon emissions and carbon sink intensity. Finally, the existing studies focus on carbon balance zoning based on carbon budget accounting, without further considering the economic contribution of carbon emissions and the ecological role of carbon sinks.

The innovative values and research contributions of this study are as follows: first, this study will improve the subdivision of land use types to more accurately account for land use carbon emissions and carbon sinks in China. Woodlands are subdivided into forest, shrubland, open woodland and other woodland, and grasslands are subdivided into high-cover grassland, medium-cover grassland and low-cover grassland. Second, carbon emissions from energy consumption are quoted with a carbon emission coefficient that is more in line with China's actual situation, which is derived from field measurements of 602 coal samples from the 100 largest coal mining areas in China. Third, this study analyzes the land use carbon balance through multiple indicators of total carbon emissions, carbon emission intensity, per capita carbon emissions, total carbon sink and carbon sink intensity, and it uses the spatial autocorrelation model to reveal the spatial differentiation characteristics of the carbon balance more deeply. Finally, considering the economic contribution coefficient (ECC) and ecological support coefficient (ESC), from the perspective of carbon balance zoning, it divides the country into four zones: low-carbon development zone, carbon sink function zone, carbon in-tensity control zone and high-carbon optimization zone, and the low carbon development path of each province is proposed in a targeted manner.

Point 2: The results/findings of this study are not discussed regarding the existing literature and more particularly regarding the pros and cons of the various proposals presented in 3.4.3. For the different zones, the authors report that they “should continue”, “should make”, “should strengthen”, “should increase”, etc. their action to cut down emissions and improve their role as sinks of carbon. But how can this be done in practice, since these policies depend upon national and regional political decisions, which are often driven by socioeconomic goals? Both discussions are missing, which is a major minus.

Response 2: Thanks for your suggestion. In this study, according to the coupling of ECC and ESC, 30 provinces in China are divided into four zones: low-carbon development zone, carbon sink function zone, carbon intensity control zone and high-carbon optimization zone, and proposes Low-Carbon development proposals. In order to verify the ideas mentioned in this paper, we add a discussion section on the proposals, as follows:

China has a large population and limited land resources, and in the rapid urbanization, the expansion of construction land is bound to squeeze other land use types, which leads to a rapid growth trend of carbon emissions [46]. Therefore, limiting the occupation of carbon sink land by construction land is significant for carbon emission reduction in China, but to meet economic development and survival needs, carbon intensity control zone and high-carbon optimization zone should increase investment in ecological environmental protection, strengthen environmental management and ecological restoration [47], optimize the layout of towns and cities, intensify land use, and build an ecological pattern of urban green areas to provide more ecological functions [48]. While woodlands and grasslands have significant carbon sink functions, low-carbon development zone and carbon sink function zone should focus on protecting ecological resources, greatly improving carbon absorption [36], and avoiding over-exploitation; most of the national ecological protection projects are distributed within these two areas [49]. China is a large carbon emitting country; hence, it is more important to reduce carbon emission sources than to increase carbon sinks [50]. The development of green industrial systems such as the service industry and high-tech industry has become the first choice [51] to ensure the quality and efficiency of economic development. The carbon intensity control zone and high-carbon optimization zone have significant responsibilities for reducing carbon emissions, and they should accelerate technological innovation [52] and upgrades by optimizing the industrial structure [13]. These measures can improve energy use efficiency and promote the transformation of resource-based cities, which will help achieve low-carbon development.

Some of the references added are as follows:

  1. Chuai, X.W.; Huang, X.J.; Lu, Q.L.; Zhang, M.; Zhao, R.Q.; Lu, J.Y. Spatiotemporal Changes of Built-Up Land Expansion and Carbon Emissions Caused by the Chinese Construction Industry. Sci. Technol. 2015, 49, 13021-13030.
  2. Han, X.Y.; Cao, T.Y.; Sun, T. Analysis on the variation rule and influencing factors of energy consumption carbon emission intensity in China's urbanization construction. Clean. Prod. 2019, 238, 117958.
  3. Chuai, X.W.; Huang, X.J.; Qi, X.X.; Li, J.S.; Zuo, T.H.; Lu, Q.L.; Li, J.B.; Wu, C.Y.; Zhao, R.Q. A Preliminary Study of the Carbon Emissions Reduction Effects of Land Use Control. Rep. 2016, 6, 36901.
  4. Huang, L.; Liu, J.Y.; Shao, Q.Q.; Xu, X.L. Carbon sequestration by forestation across China: Past, present, and future. Sust. Energ. Rev. 2011, 16, 1291-1299.
  5. Chen, J.D.; Li, Z.W.; Song, M.L.; Dong, Y.Z. Decomposing the global carbon balance pressure index: evidence from 77 countries. Sci. Pollut. Res. Int. 2020, 28, 7016-7031.
  6. Xia, F.; Yang, Y.X.; Zhang, S.Q.; Li, D.H.; Sun, W.; Xie, Y.J. Influencing factors of the supply-demand relationships of carbon sequestration and grain provision in China: Does land use matter the most? Total Environ. 2022, 832, 154979.
  7. Li, L.; Hong, X.F.; Peng, K. A spatial panel analysis of carbon emissions, economic growth and high-technology industry in China. Chang. Econ. Dyn. 2019, 49, 83-92.

Point 3: The overall number of references (31) is low. Consequently, the theoretical background is poor and the discussion in dialogue with the literature is missing. Please, try to improve both sections. There is a profuse number of studies conducted in China that could be referenced in this manuscript.

Response 3: Thanks for the references and suggestions, we have added a discussion section according to your advice. First, we have added literature citations in order to enhance the theoretical background. Second, we have fully discussed the findings of this study in terms of land use intensity, carbon emissions and carbon sinks, as follows:

In terms of land use pattern and intensity, China exhibits strong land use intensity, with high values concentrated in the Bohai Rim and the eastern coastal region and low values mainly distributed in the western region. which is similar to the findings of Chen et al. [35]. The area of carbon sink land in China is much larger than that of carbon source land, and the land use types are mainly grassland, woodland, cultivated land, and unused land [36]. Among them, carbon source land is mainly cultivated land, followed by construction land, which is in line with the reality of a large agricultural country and the continuous expansion of construction land in China [37,38], while carbon sink land is mainly woodland, grassland and unused land.

In recent years, with the rapid development of China's economy, total carbon emissions have continued to increase and have been confirmed by the academic community [39,40]. Jing et al found that the total carbon emissions in China were 109.5×108 t in 2010 and 138.5×108 t in 2019, with an average annual growth rate of 2.37% [41]. In this study, the total carbon emissions in China in 2018 were 114.47×108 t, which is still considered high, and this value is still in a reasonable range compared with previous studies and is more accurate than the findings of previous studies, which consider the apparent energy consumption (raw coal, crude oil and natural gas) and use a carbon emission coefficient that is more in line with China's actual situation. Construction land is the main source of carbon emissions in China and cultivated land accounts for only a very small portion, which is consistent with previous research results [41,42]. The aggregation effect of carbon emission intensity in China is obvious, and in general, it shows a scattered distribution of high values, an aggregated distribution of low values and a decreasing trend from west to the east and from north to south [43]. The total national carbon sink is 130.76×106 t, and woodlands and grasslands account for the majority of the carbon sink; woodlands and grasslands are the main carriers of terrestrial carbon absorption in China [44,45] and have the most obvious offsetting effect on carbon emissions.

Some of the references added are as follows:

  1. Chen, W.X.; Zeng, J. Decoupling analysis of land use intensity and ecosystem services intensity in China. Nat. Resour. 2021, 36, 2853-2864.
  2. Li, J.S.; Guo, X.M.; Chuai, X.W.; Xie, F.J.; Yang, F. Reexamine China’s terrestrial ecosystem carbon balance under land use-type and climate change. Land Use Policy. 2021, 102, 105275.
  3. Tang, X.L.; Lu, C.Y.; Meng, P.; Cheng, W. Spatiotemporal Evolution of the Environmental Adaptability Efficiency of the Agricultural System in China. Sustainability. 2022, 14, 3685.
  4. Zhu, M.C.; Shen, L.Y.; Tam, V.W.Y.; Liu, Z.; Shu, T.H.; Luo, W.Z. A load-carrier perspective examination on the change of ecological environment carrying capacity during urbanization process in China. Total Environ. 2020, 714, 136843.
  5. Xu, W.H.; Xie, Y.L.; Ji, L.; Cai, Y.P.; Yang, Z.F.; Xia, D.H. Spatial-temporal evolution and driving forces of provincial carbon footprints in China: An integrated EE-MRIO and WA-SDA approach. Eng. 2022, 176, 106543.
  6. Lin, Q.W.; Zhang, L.; Qiu, B.K.; Zhao, Y.; Wei, C. Spatiotemporal Analysis of Land Use Patterns on Carbon Emissions in China. Land. 2021, 10, 141.
  7. Jing, X.D.; Tian, G.L.; Li, M.R.; Javeed, S.A. Research on the Spatial and Temporal Differences of China’s Provincial Carbon Emissions and Ecological Compensation Based on Land Carbon Budget Accounting. J. Environ. Res. Pub He. 2021, 18, 12892.
  8. Huang, H.Q.; Zhou, J. Study on the Spatial and Temporal Differentiation Pattern of Carbon Emission and Carbon Compensation in China’s Provincial Areas. Sustainability. 2022, 14, 7627.
  9. Yang, M.; Liu, Y.S.; Tian, J.Z.; Cheng, F.Y.; Song, P.B. Dynamic Evolution and Regional Disparity in Carbon Emission Intensity in China. Sustainability. 2022, 14, 4052.
  10. Wang, Y.L.; Wang, X.H.; Wang, K.; Chevallier, F.; Zhu, D.; Lian, J.H.; He, Y.; Tian, H.Q.; Li, J.S.; Zhu, J.X.; Jeong, S.J.; Canadell, J.G. The size of the land carbon sink in China. Nature. 2022, 603, E7-E9.
  11. Wang, J.; Feng, L.; Palmer, P.I.; Liu, Y.; Fang, S.X.; Bösch, H.; O'Dell, C.W.; Tang, X.P.; Yang, D.G.; Liu, L.X.; Xia, C.Z. Large Chinese land carbon sink estimated from atmospheric carbon dioxide data. Nature. 2020, 586, 720-723.

Point 4: In some parts, this manuscript has a structure and a content similar to the paper published by Wang et al. , 2021 (reference [13]). For example, Table 1 and Figure 3 seems to be clearly retrieved from that publication. Although the content is not exactly the same, the authors should avoid this type of approach to avoid plagiarism issues.

Response 4: We apologize for causing you doubts. This article was conceived and written by reviewing a large amount of literature to better conduct the research. To make the presentation of data in the article more intuitive, we used software such as ArcGIS, GeoDa and Adobe Illustrator to create figures. The Table 1 and Figure 3 you mentioned borrowed from the paper published by Wang et al., we considered that the presentation of the figure is suitable for this study, which leads to similar problems. However, the structure and content of this article were determined based on our thorough discussions, and we will be more cautious when conducting future articles. Again, we sincerely thank you for your reminder.

Additional comments

Line 31: The abbreviation “LUCC” is not necessary since it is not used in the text.

It could be interesting to mention the software used to create the maps of Figures 2 and 5 to 8.

Figure 2: It would be interesting to identify and delimit the four main regions (eastern region, central region, western region and northeastern region) on the map of Figure 2, by using for example dissolved polygons with a larger borderline. Is this map representing the 30 provinces mentioned in the abstract? Data used to build the map of Figure 2 (and overall maps) should be described in the respective source, which is missing.

Line 119: Please provide a reference for the “China statistical yearbook”.

Figure 3: In the arrow, the term “National region” is not very appropriated, please consider to change it.

Figure 5: These legends are very difficult to read, because the maps are too small. Please try to make these legends larger.

Some sentences need to be revise. For example, in line 64 (“Third, studies on carbon balance, zoning and optimization schemes.”) something is apparently missing

Response 5: Thanks for your careful check. responses to additional comments are as follows:

  1. We have removed the LUCC abbreviation.
  2. In order to identify the four main regions, we have added a figure on the four regions (Figure 2. Four regions geographical divisions of China), and the figure has included the 30 provinces mentioned in the abstract and the map source situation has been annotated in Figure 2.
  3. We have supplemented the reference of "China Statistical Yearbook" with the reference number 22.
  4. We have changed "National region" to "National scale".
  5. All figures and tables in the article have been carefully checked and revised to ensure that the information is presented visually to the viewer.
  6. We have rechecked and corrected the presentation of the article.

We look forward to hearing from you regarding our submission. We would be glad to respond to any further questions and comments that you may have.

Sincerely,

Authors

23th September, 2022

Reviewer 3 Report

This work analysis the spatial characteristics of carbon emissions and carbon sinks in 30 Chinese provinces and the carbon balance by combining the economic contribution coefficient (ECC) and ecological support coefficient (ESC). This research is of great significance to curb global warming. Personally, I think the authors have done a lot of work, however, there are some deficiencies that need to be further improved.

Question1#: The authors show that most research focuses on the regional and city/county scales, and studies at the national scale are still rare. As far as I know, large scale such as global, national research are relatively more, while small scale, fine scale research or accurate measurement, etc. are relatively less.

Question2#: The overall logic of the introduction part is relatively lacking, and it needs to be further condensed and summarized. For example, it needs to highlight the research contribution of this article, the innovation value of this article.

Question3#: Line 225 (part 3.3), it may not be enough to explain the spatial differentiation only from the spatial distribution characteristics, which may need to be analyzed by the Theil index, LISA or other indexes.

Question4#: Reference format needs to be revised according to journal format, such as abbreviations, bolding, punctuation, etc.

Author Response

Response to Reviewer 3 Comments

Dear Reviewer:

We highly appreciate the reviewer for the insightful comments and criticism, which have helped us improve both the content and the presentation of our work. We believe that the revised version of our paper addresses all concerns by the referees in detail. The changes are highlighted within the manuscript. Please see below, in red, for a point-by-point response to the reviewer’s comments and concerns.

Point 1: The authors show that most research focuses on the regional and city/county scales, and studies at the national scale are still rare. As far as I know, large scale such as global, national research are relatively more, while small scale, fine scale research or accurate measurement, etc. are relatively less.

Response 1: We apologize for causing you doubts due to some of our poor presentation, and we have rechecked and corrected the presentation of the article. Regarding the issue of the research scale you mentioned, the gaps in the existing literature and the innovations and contributions of this paper are condensed and summarized in the introduction part.

Point 2: The overall logic of the introduction part is relatively lacking, and it needs to be further condensed and summarized. For example, it needs to highlight the research contribution of this article, the innovation value of this article.

Response 2: We sincerely appreciate the reviewer’s question. We have reorganized the logic of the introduction part and further condensed and summarized it to highlight the research contribution and innovative value of this article, as follows:

In general, there have been systematic studies on carbon emissions, carbon sinks and the land use carbon balance, which are important references for this work; however, the following problems are still worth exploring in depth. First, the subdivision of land use types is not deep enough, which leads to large errors in the measurement results of the land use carbon budget. Second, most of the carbon emission factors for energy consumption use data published by the IPCC; however, China, as a large energy-consuming country, needs to determine its carbon emission factors through field measurements. Third, most scholars analyze the land use carbon budget from two perspectives, total carbon emissions and total carbon sinks, but fail to study it from multiple perspectives, such as carbon emission intensity, per capita carbon emissions and carbon sink intensity. Finally, the existing studies focus on carbon balance zoning based on carbon budget accounting, without further considering the economic contribution of carbon emissions and the ecological role of carbon sinks.

The innovative values and research contributions of this study are as follows: first, this study will improve the subdivision of land use types to more accurately account for land use carbon emissions and carbon sinks in China. Woodlands are subdivided into forest, shrubland, open woodland and other woodland, and grasslands are subdivided into high-cover grassland, medium-cover grassland and low-cover grassland. Second, carbon emissions from energy consumption are quoted with a carbon emission coefficient that is more in line with China's actual situation, which is derived from field measurements of 602 coal samples from the 100 largest coal mining areas in China. Third, this study analyzes the land use carbon balance through multiple indicators of total carbon emissions, carbon emission intensity, per capita carbon emissions, total carbon sink and carbon sink intensity, and it uses the spatial autocorrelation model to reveal the spatial differentiation characteristics of the carbon balance more deeply. Finally, considering the economic contribution coefficient (ECC) and ecological support coefficient (ESC), from the perspective of carbon balance zoning, it divides the country into four zones: low-carbon development zone, carbon sink function zone, carbon in-tensity control zone and high-carbon optimization zone, and the low carbon development path of each province is proposed in a targeted manner.

Point 3: Line 225 (part 3.3), it may not be enough to explain the spatial differentiation only from the spatial distribution characteristics, which may need to be analyzed by the Theil index, LISA or other indexes.

Response 3: Thanks for your suggestion. In order to explain the spatial differentiation of carbon emissions and carbon sinks in more depth, we have conducted a thorough discussion based on your suggestions, used a spatial autocorrelation model to further analyze the spatial distribution characteristics (part 3.3.3), and drew a LISA spatial clustering pattern map (Figure 8, Figure 9). As follows:

3.3.3. Spatial autocorrelation analysis

According to Tobler's First Law, spatial adjacency factors influence the spatial distribution of carbon emissions and carbon sinks to a certain extent. The global spatial autocorrelation results show that the global Moran's I of total carbon emissions, carbon emission intensity, per capita carbon emissions, total carbon sink and carbon sink intensity are 0.191, 0.268, 0.277, 0.286 and 0.559, respectively, indicating a significant positive spatial correlation and spatial aggregation characteristics, among which the positive correlation effect of carbon sink intensity is the strongest.

To further study the local spatial correlation between carbon emissions and carbon sinks, Local Indicators of Spatial Association (LISA) spatial distribution maps were drawn using GeoDa and ArcGIS (Figure 8, Figure 9). It can be found that carbon emissions and carbon sinks mainly show H-H clusters and L-L clusters, in which the local aggregation effects of carbon emission intensity, per capita carbon emissions and carbon sink intensity are the most significant. In terms of total carbon emissions (Figure 8a), H-H types were distributed in Inner Mongolia, Shanxi, Henan and Hebei, where total carbon emissions were high near the region; L-L types were distributed in Sichuan, Guizhou and Hunan. In terms of carbon emission intensity (Figure 8b), L-L types mainly distributed in southeastern coastal and central provinces, and the occurrence of H-L and L-H types indicates the obvious heterogeneity of these provinces. In terms of per capita carbon emissions (Figure 8c), H-H types were distributed in five provinces, Inner Mongolia, Ningxia, Shaanxi, Shanxi & Liaoning; L-L types were mainly distributed in southern China, and no H-L types were present. In terms of the total carbon sink (Figure 9a), H-H clusters are distributed in provinces with abundant forest resources, and L-H types were distributed in Guizhou & Liaoning, and no H-L types existed. In terms of carbon sink intensity (Figure 9b), H-H clusters distributed in six provinces in southern China, L-L clusters distributed in provinces north of the Qinling Mountains-Huaihe River Line, with obvious local spatial correlation effects.

Point 4: Reference format needs to be revised according to journal format, such as abbreviations, bolding, punctuation, etc.

Response 4: Thanks for your careful check. We have carefully revised the reference format according to the journal format to ensure compliance with the journal requirements.

We look forward to hearing from you regarding our submission. We would be glad to respond to any further questions and comments that you may have.

Sincerely,

Authors

23th September, 2022

Round 2

Reviewer 2 Report

Dear authors,

Please find in the attached file my comments to this revised version of your manuscript. Thank you.

Author Response

Response to Reviewer 2 Comments

Dear Reviewer:

Thank you for your review of the manuscript. We highly appreciate the reviewer for the encouraging and enthusiastic comments and suggestions, on the basis of which we have revised and strengthened our paper. We believe that the revised version of our paper addresses all concerns by the referees in detail. The changes are highlighted within the manuscript. Please see below, in red, for a point-by-point response to the reviewer’s comments and concerns.

Point 1: Consider to specify the number of land use types analyzed in this study. It is also important to clarify this issue on the main manuscript, because it is unclear how many land use types/categories was analyses: 13 types (6 main categories + 7 subtypes as found in lines 119-123) or 11 types as shown in Figure 4?

Response 1: We sincerely appreciate the reviewer’s question. In order to specify the number of land use types analyzed in this study, while ensuring the brevity of the abstract, we chose to clarified the issue in part 2.1. A total of 11 land use types were analyzed in this paper, as follows:

Land use types were divided into the following categories: cultivated land, woodland, grassland, waterbody, construction land and unused land, among which woodland was subdivided into forest, shrubland, open woodland and other woodland, and grass-land was subdivided into high-cover grassland, medium-cover grassland and low-cover grassland. Therefore, land use types were divided into 11 categories in total.

Point 2: Keywords: Consider to include “Carbon emissions”.

Response 2: Thanks for your suggestion. We have included “Carbon emissions” as a keyword.

Point 3: Lines 27-29: I think the target “to strive to achieve carbon peak by 2030 and carbon neutrality by 2060” was the China's commitment to the Paris Agreement. Please, consider to mention the Paris Agreement on this sentence and in other parts of this manuscript.

Response 3: Thanks for your suggestion. We have added the Paris Agreement to this sentence and the introduction as follows:

Global warming caused by the rapid increase in carbon emissions has become a major focus of attention, and the promotion of green and low-carbon economic development has become the consensus strategy among the international community to address and improve climate change [1]. Therefore, the milestone international legal text, the Paris Agreement, has been successfully implemented, making arrangements for global action against climate change after 2020 and forming a new pattern of global climate governance. China became the world's top carbon emitter in 2009 and accounted for 23.87% of the world's total carbon emissions in 2017 [2]. In order to strengthen China's Nationally Determined Contributions and implement the commitments of the Paris Agreement, China has pledged to strive to achieve carbon peak by 2030 and carbon neutrality by 2060.

Point 4: Line 72: Please provide examples of those “important references for this work”. Otherwise, the sentence remains vague.

Response 4: We have provided examples of "important references for this work" with the reference numbers 8, 12, 13, 19.

Point 5: Line 76: Please describe the meaning of “IPCC”. Check if the meaning of all abbreviations is given in the entire manuscript.

Response 5: We have checked the entire manuscript for all abbreviations to ensure that the meaning of the abbreviations is clear, and the meaning of IPCC is described as Intergovernmental Panel on Climate Change.

Point 6: Line 84: In my view, this sentence should start as follows: “Therefore, to fulfil the aforementioned gaps, this study analyzes the land use status and intensity of each Chinese province to understand…”

Response 6: Thanks for your suggestion. After our thorough discussion, we have revised this paragraph, as follows:

Therefore, to fulfil the aforementioned gaps, this study analyzes the land use status and intensity of each Chinese province to understand the pattern of land use carbon sources/sinks as a whole. Secondly, it measures the total amount of carbon emissions and carbon sinks and analyzes their spatial distribution. Finally, it integrates the ECC and ESC of carbon emissions into the carbon balance zoning of the country, and differentiated low-carbon optimization suggestions are put forward according to local conditions, aiming to provide a theoretical basis for relevant departments to adjust land use policies and improve the distribution efficiency of carbon emission reduction tasks in each province.

Point 7: Lines 114-115: This sentence explains that “data used in this study mainly include land use, energy consumption, and socioeconomic data”, but the subsequent description has four items, not three (Carbon emission and carbon sequestration coefficients are not reported in lines 114-115…).

Response 7: Thanks for your careful check. To reflect the logic of this paragraph, we have completed the opening sentence, as follows:

The data used in this study mainly include land use, energy consumption, socioeconomic and carbon emission and carbon sequestration coefficients data of 30 provincial-level administrative regions in China (excluding Tibet, Hong Kong, Macao, and Taiwan).

Point 8: Line 239: Please consider to change “showing a trend of ups and downs” by a more appropriated expression.

Response 8: Thanks for your suggestion. We have changed "showing a trend of ups and downs" to "showing a tortuous and fluctuating trend".

Point 9: Line 281: Please try to be more objective since “rich tourism resources” could mean many different resources (built heritage, intangible heritage, nature, etc.).

Response 9: Thanks for your careful check. This article is mainly to analyzed the carbon emissions and carbon sinks of Chinese provinces, so the sentence wants to emphasize the natural tourism resources. We have replaced "rich tourism resources" with "rich natural tourism resources".

Point 10: Line 309: Are these “10.02 t/person” the average amount produced per year?

Response 10: The 10.02 t/person represents the average production per person per year. This paper analyzed the total carbon emissions of each province in China in 2018, and the per capita carbon emissions are obtained by dividing the total carbon emissions by the total population. Therefore, the meaning of the unit of per capita carbon emissions is the average production per person per year.

Point 11: Lines 344-349: Very longue sentence. Consider to divide it.

Response 11: Thanks for your suggestion. In order to make this sentence clearer, after our discussion, this sentence has been arranged and modified as follows:

The total amount of carbon sinks in most provinces in China is below 5.00106 t, and the function of carbon sinks is weak. In the process of urbanization, the construction land is expanding rapidly, which constricts the living space of carbon sink land.

Point 12: Line 550: Please replace the full stop before “which” by a comma.

Response 12: Thanks for your careful check. We have changed it to a comma.

Point 13: Line 563: Please add examples of those “previous studies”.

Response 13: We have provided examples of "previous studies" with the reference numbers 39,41.

Point 14: Line 587: It could be interesting to start the discussion by introducing this sentence: “China is a large carbon emitting country; hence, it is more important to reduce carbon emission sources than to increase carbon sinks [50].” Then, try to arrange the discussion around this main idea.

Response 14: Thanks for your suggestion. After our full discussion, we have revised the discussion section and added two references, as follows:

China is a large carbon emitting country; hence, it is more important to reduce carbon emission sources than to increase carbon sinks [46]. China has a large population and limited land resources, and in the rapid urbanization, the expansion of construction land is bound to squeeze other land use types, which leads to a rapid growth trend of carbon emissions [47]. Therefore, limiting the occupation of carbon sink land by construction land is significant for carbon emission reduction in China. In terms of industrial system, the development of green industries such as the service industry and high-tech industries has become the first choice [48]. The carbon intensity control zone and high-carbon optimization zone have significant responsibilities for reducing carbon emissions, and they should accelerate technological innovation and upgrades [49], and optimizing the industrial structure [13]. These measures can improve energy use efficiency and promote the transformation of resource-based cities, which will help achieve low-carbon development. To achieve carbon neutrality in China, it is very difficult to rely solely on the increase of carbon sequestration. However, this doesn't mean that ecological protection and afforestation should be neglected. A good natural environment is not only the largest contributor to carbon sinks [50], but also provides a variety of ecosystem service functions such as soil and water conservation and conservation of species diversity [51]. Therefore, the low-carbon development zone and carbon sink function zone should focus on protecting ecological resources, greatly improving carbon absorption [36], and avoiding overexploitation; most of the national ecological protection projects are distributed within these two areas [52]. The carbon intensity control zone and high-carbon optimization zone should increase investment in ecological environmental protection, strengthen environmental management and ecological restoration [53], optimize the layout of towns and cities, intensify land use, and build an ecological pattern of urban green areas to provide more ecological functions [54].

Some of the references added are as follows:

  1. Fang, J.Y.; Guo, Z.D.; Hu, H.F.; Kato, T.; Muraoka, H.; Son, Y. Forest biomass carbon sinks in East Asia, with special reference to the relative contributions of forest expansion and forest growth. Chang. Biol. 2014, 20, 2019-2030.
  2. Jonsson, M.; Bengtsson, J.; Gamfeldt, L.; Moen, J.; Snäll, T. Levels of forest ecosystem services depend on specific mixtures of commercial tree species. plants. 2019, 5, 141-147.

Point 15: The terms “north”, “south”, “east” and “west” should be capitalized when they are part of a proper noun or when they designate formal regions. Moreover, in some sentences, directions appear capitalized (such as “Northeastern” in line 209), while in other sentences the same word is not capitalized (line 203 for example). Try to avoid this type of inconsistencies.

Response 15: Thanks for your careful check. We have checked the entire manuscript to avoid this type of inconsistencies. When “north”, “south”, “east” and “west” stand for four regions, we have capitalized them.

We look forward to hearing from you regarding our submission. We would be glad to respond to any further questions and comments that you may have.

Sincerely,

Authors

2th October, 2022

Reviewer 3 Report

This version is significantly improved compared with the previous version. There are still some small language problems that can be slightly changed. In addition, if the results of this paper can be compared with the results of previous studies, it will be more perfect.

Author Response

Response to Reviewer 3 Comments

Dear Reviewer:

Thank you for your review of the manuscript. We highly appreciate the reviewer for the encouraging and enthusiastic comments and suggestions, on the basis of which we have tried our best to revise our manuscript. We believe that the revised version of our paper addresses all concerns by the referees in detail. The changes are highlighted within the manuscript. Please see below, in red, for a point-by-point response to the reviewer’s comments and concerns.

Point 1: There are still some small language problems that can be slightly changed.

Response 1: Thanks for your careful check. We have checked the entire manuscript to ensure fluency and clarity of language.

Point 2: In addition, if the results of this paper can be compared with the results of previous studies, it will be more perfect.

Response 2: Thanks for your suggestion. After our thorough discussion, we tried our best to compared the results of this paper with previous studies in the discussion section, and we have organized and revised the discussion section and added references, as follows:

China is a large carbon emitting country; hence, it is more important to reduce carbon emission sources than to increase carbon sinks [46]. China has a large population and limited land resources, and in the rapid urbanization, the expansion of construction land is bound to squeeze other land use types, which leads to a rapid growth trend of carbon emissions [47]. Therefore, limiting the occupation of carbon sink land by construction land is significant for carbon emission reduction in China. In terms of industrial system, the development of green industries such as the service industry and high-tech industries has become the first choice [48]. The carbon intensity control zone and high-carbon optimization zone have significant responsibilities for reducing carbon emissions, and they should accelerate technological innovation and upgrades [49], and optimizing the industrial structure [13]. These measures can improve energy use efficiency and promote the transformation of resource-based cities, which will help achieve low-carbon development. To achieve carbon neutrality in China, it is very difficult to rely solely on the increase of carbon sequestration. However, this doesn't mean that ecological protection and afforestation should be neglected. A good natural environment is not only the largest contributor to carbon sinks [50], but also provides a variety of ecosystem service functions such as soil and water conservation and conservation of species diversity [51]. Therefore, the low-carbon development zone and carbon sink function zone should focus on protecting ecological resources, greatly improving carbon absorption [36], and avoiding overexploitation; most of the national ecological protection projects are distributed within these two areas [52]. The carbon intensity control zone and high-carbon optimization zone should increase investment in ecological environmental protection, strengthen environmental management and ecological restoration [53], optimize the layout of towns and cities, intensify land use, and build an ecological pattern of urban green areas to provide more ecological functions [54].

Some of the references added are as follows:

  1. Fang, J.Y.; Guo, Z.D.; Hu, H.F.; Kato, T.; Muraoka, H.; Son, Y. Forest biomass carbon sinks in East Asia, with special reference to the relative contributions of forest expansion and forest growth. Chang. Biol. 2014, 20, 2019-2030.
  2. Jonsson, M.; Bengtsson, J.; Gamfeldt, L.; Moen, J.; Snäll, T. Levels of forest ecosystem services depend on specific mixtures of commercial tree species. plants. 2019, 5, 141-147.

We look forward to hearing from you regarding our submission. We would be glad to respond to any further questions and comments that you may have.

Sincerely,

Authors

2th October, 2022